# Somatic *PIK3R1* mutations in the iSH2 domain are accessible to PI3Kα inhibition

Gabriel Morin [1,2,3,19], Alexandre P Garneau [1,2,19], Nabiha Bouzakher[1,2], Louise Ségot[4], Antoine Fraissenon [2,5,6,7], Amélie Blondel[1,2], Florence Petit[8], Caroline Chopinet[9], Franck Mayeux[1,2], Pierre Fayoux[10], Anne Dompmartin[11], Christine Bodemer[1,12], Estelle Balducci[1,13], Sophie Kaltenbach[1,13], Patrick Villarese[13], Vahid Asnafi[1,2,13], Christophe Legendre[1,14], Christine Broissand[15], Sylvie Fraitag[16], Chloé Quelin[17], Nicolas Goudin [18], Laurent Guibaud[2,5] & Guillaume Canaud [1,2,3 ✉]

## Abstract

Mutations in *PIK3R1* have recently been identified in patients with overgrowth syndromes and complex vascular malformations. *PIK3R1* encodes p85α which acts as the regulatory subunit of the lipid kinase PI3Kα. *PIK3R1* mutations result in the excessive activation of the AKT/mTOR pathway. Currently, there are no approved treatments specifically dedicated to patients with *PIK3R1* mutations, and medical care primarily focuses on managing symptoms. In this study, we identified three patients, including two children, who had mosaic somatic *PIK3R1* mutations affecting the iSH2 domain, along with severe associated symptoms that were unsuccessfully treated with rapamycin. We conducted in vitro experiments to investigate the impact of these mutations, including a double *PIK3R1* mutation in cis observed in one patient. Our findings revealed that p85α mutants in the iSH2 domain showed sensitivity to alpelisib, a pharmacological inhibitor of PI3Kα. Based on these findings, we received authorization to administer alpelisib to all three patients. Following drug introduction, patients rapidly demonstrated clinical improvement, pain, fatigue and inflammatory flares were attenuated. Magnetic Resonance Imaging showed a mean decrease of 22.67% in the volume of vascular malformations over twelve months of treatment with alpelisib. No drug-related adverse events were reported during the course of the study. In conclusion, this study provides support for the use of PI3Kα inhibition as a promising therapeutic approach for individuals with *PIK3R1*-related anomalies.

**Keywords** PIK3R1-Related Disorders; p85; Vascular Malformations; Overgrowth Syndrome; Alpelisib
**Subject Categories** Cardiovascular System; Vascular Biology & Angiogenesis

## Introduction

Overgrowth syndromes encompass rare genetic disorders characterized by the abnormal growth of tissues, which can occur either locally or throughout the body, affecting both width and length. Typically, these mutations arise during embryonic development, resulting in somatic mosaicism (Canaud et al, 2021). In most cases, overgrowth syndromes are caused by somatic gain-of-function mutations in the PI3Kα complex (Canaud et al, 2021). The PI3Kα complex is a lipid kinase complex composed of a catalytic subunit called p110α, encoded by the *PIK3CA* gene, and a regulatory subunit known as p85α, encoded by the *PIK3R1* gene. The p85α subunit binds to and inhibits p110α (Madsen and Vanhaesebroeck, 2020). p85α consists of several domains, including a Src homology region 3 (SH3), a breakpoint cluster region homology (BH) domain with GTPase activating protein (GAP) activity, and a Src homology region 2 domain (SH2) with N-terminal (nSH2), inter (iSH2), and C-terminal (cSH2) SH2 domains (Backer, 2010). The first two domains bind to the phosphatase and tensin homolog (PTEN) (Cheung et al, 2011). Upon activation by growth factors, the nSH2 and cSH2 domains of p85α bind to phosphorylated tyrosine residues in activated receptors and adapters, thereby activating the catalytic p110α (Backer, 2010). P110α converts, at the plasma membrane, phosphatidylinositol 4,5-bisphosphate (PtdIns(4,5)P2) to phosphatidylinositol 3,4,5-trisphosphate (PtdIns(3,4,5)P3; or PIP3) which subsequently recruits phosphoinositide-dependent protein kinase 1 (PDK1). PDK1 then phosphorylates AKT on the

[1]Université Paris Cité, Paris, France. [2]INSERM U1151, Institut Necker-Enfants Malades, Paris, France. [3]Unité de Médecine Translationnelle et Thérapies Ciblées, Hôpital Necker-Enfants Malades, AP-HP, Paris, France. [4]Polytech Nice Sophia, Université Côte d'Azur, Nice, France. [5]Service d'Imagerie Pédiatrique, Hôpital Femme-Mère-Enfant, HCL, Bron, France. [6]CREATIS UMR 5220, Villeurbanne 69100, France. [7]Service de Radiologie Mère-Enfant, Hôpital Nord, Saint Etienne, France. [8]Clinique de Génétique, CHU de Lille, Lille 59000, France. [9]Service de Physiologie & Explorations Fonctionnelles Cardiovasculaires, CHU de Lille, Lille 59000, France. [10]Service d'ORL, CHU de Lille, Lille 59000, France. [11]Service de Dermatologie, CHU Côte de Nacre, Caen 14033, France. [12]Service de Dermatologie, Hôpital Necker-Enfants Malades, AP-HP, Paris, France. [13]Laboratoire d'Oncohématologie, Hôpital Necker-Enfants Malades, AP-HP, Paris, France. [14]Service de Néphrologie, Transplantation Adultes, Hôpital Necker-Enfants Malades, AP-HP, Paris, France. [15]Pharmacie, Hôpital Necker-Enfants Malades, AP-HP, Paris, France. [16]Service d'Anatomie Pathologique, Hôpital Necker-Enfants Malades, AP-HP, Paris, France. [17]Service de Génétique Clinique, Centre de Référence Anomalies du Développement de l'Ouest, CHU Rennes, Rennes, France. [18]Structure Fédérative de Recherche Necker, INSERM US24,-CNRS UAR 3633, Institut Necker-Enfants Malades, Paris, France. [19]These authors contributed equally: Gabriel Morin, Alexandre P Garneau. ✉E-mail: guillaume.canaud@inserm.fr

Thr308 residue, initiating downstream cellular effects associated with cell proliferation, motility, survival, and metabolism (Backer, 2010). P85α also regulates the PI3Kα pathway by stabilizing the phosphatase PTEN (Cheung et al, 2011). While most cases of overgrowth syndromes can be explained by gain-of-function mutations in *PIK3CA*, recent studies have reported *PIK3R1* mutations in some patients with overgrowth syndromes and vascular malformations (Cottrell et al, 2021). In oncology, hotspot mutations in *PIK3R1* are often found between the nSH2, iSH2, and cSH2 domains and have been shown to increase PI3Kα activity (Liu et al, 2014). In previous research, we identified alpelisib as a promising pharmacological inhibitor of PI3Kα for patients with *PIK3CA*-related overgrowth spectrum (PROS), which led to its recent approval by the US FDA (Bayard et al, 2023; Canaud et al, 2023a; Delestre et al, 2021; Ladraa et al, 2022; Morin et al, 2022; Venot et al, 2018; Zerbib et al, 2024). Since the phenotypic spectrum of patients with *PIK3R1* mutations overlaps with that of PROS patients with similar underlying mechanisms and the recent encouraging results obtained with alpelisib in a pediatric patient carrying a *PIK3R1* mutation (Schonewolf-Greulich et al, 2022), we investigated the impact of alpelisib in three patients with somatic *PIK3R1* mutations.

## Results

### PIK3R1 mutations in the iSH2 domain activate the PI3Kα pathway in vitro and are accessible to alpelisib

We identified 3 patients with somatic *PIK3R1* mutations and severe associated disorders (see below for the complete cases description). Patient 1 had a *PIK3R1 c.1735_1740del* variant (VAF 18%, ACMG class 5, pathogenic) (p.[Q579-Y580]del), patient 2 had a *PIK3R1 c.1372_1373dupAAA* variant (VAF 15%, ACMG class 5, pathogenic) (p.K459dup) and patient 3 had a double *PIK3R1* mutation including *c.1699A>G* (VAF 8%, ACMG class 5, pathogenic) and *c.1703C>T* variants (VAF 8% ACMG class 4, likely pathogenic) in cis (p.[K567E,P568L]). These 3 variants localize in the iSH2 domain of p85α (p.[Q579-Y580]del, p.K459dup and p.[K567E,P568L], respectively) (Cheung and Mills, 2016).

To begin, we examined skin biopsies obtained from all three patients to determine whether the AKT/mTORC1 pathway was activated. By immunofluorescence, we observed that AKT was activated in the skin biopsies of all three patients, while mTORC1 was not, in comparison to the control group of healthy individuals (Fig. 1A). Subsequently, we studied the effects of *PIK3R1* mutations on the PI3Kα pathway in vitro. To this aim, we transfected HeLa cells with plasmids encoding GFP, wild-type *PIK3R1* (*PIK3R1^WT^*) or *PIK3R1* variants (*PIK3R1^c.1735_1740del^*, *PIK3R1^c.1372_1373dupAAA^*, and *PIK3R1^c.1699A>G, c.1703C>T^*).

We initially observed that transfection of HeLa cells with the first variant, *c.1735_1740del*, resulted in AKT phosphorylation without a concurrent increase in S6RP phosphorylation (Fig. 1B). Notably, the phosphorylation of AKT was effectively attenuated by alpelisib treatment (Fig. 1B). The second patient's variant (*PIK3R1 c.1372_1373dupAAA*) led to the activation of AKT but not to the phosphorylation of S6RP (Fig. 1B). Treatment with alpelisib effectively inhibited AKT phosphorylation in this case (Fig. 1B).

Furthermore, we transfected HeLa cells with double *c.1699A>G* and *c.1703C>T* variants in cis, which resulted in an increase in AKT phosphorylation that was suppressed by alpelisib (Fig. 1C). Interestingly, the double mutant in cis led to a stronger activation of AKT compared to single mutants (Fig. EV1).

Given the effectiveness of alpelisib at dampening the phosphorylation of AKT induced by *PIK3R1* variants, we hypothesized that p85α mutants should be able to bind p110α to dimerize into PI3Kα despite the amino-acid changes in the iSH2 domain induced by the mutations. We first wondered whether the activation of AKT we observed in cells transfected with *PIK3R1* variants could be the consequence of changes in the expression levels of p110α. As expected, p110α protein expression remained similar in HeLa cells regardless of the *PIK3R1* variant transfected (Fig. 2A,B). We then assessed whether the *PIK3R1* variants led to conformational changes in the PI3Kα heterodimer. To do so, we performed in silico modeling of wild-type and mutant PI3Kα for all three *PIK3R1* variants with AlphaFold3. The 3D models predicted had high accuracy (ipTM≥0.88) and the position of the SH2 domains regarding p110α domains showed very low expected position errors (Fig. EV2), thereby suggesting reliable predicted structures. We next computed the RMSD values of the 3 predicted variant and observed that for all three *PIK3R1* variants, values were very low (from 0.095 to 0.27), suggesting that wild-type and mutant PI3Kα have similar conformations (Fig. EV2). To further confirm that p85α mutants interact with p110α despite changes in their iSH2 domains, we performed co-immunofluorescence experiments in HeLa cells transfected with *PIK3R1* variants. We observed that the p85α-positive fraction of the cell volume colocalizing with p110α was similar in cells transfected with the two first variants *c.1735_1740del* (10.9% ± 3.5, ns) and *c.1372_1373dupAAA* (14.4% ± 6.1, ns) compared to cells transfected with wild-type *PIK3R1* (11.6% ± 3.7). The colocalization volume was even increased in cells expressing the double mutant *c.1699A>G, c.1703C>T* (24.6% ± 4.8, $p < 0.0001$) (Fig. 2C). Considering the efficacy of alpelisib at decreasing AKT phosphorylation, the bioinformatic predictions and the colocalization results, we concluded that p85α mutants signal through PI3Kα and that activation of AKT is p110α-dependent.

Since we observed that rapamycin was ineffective in the 3 patients and that the level of phosphorylation of S6RP was not modified by the *PIK3R1* variants, we investigated the phosphorylation of potential downstream AKT substrates in HeLa cells transfected with the variants and appropriate controls. We observed that the 3 variants produced a similar pattern of phosphorylated proteins, indicating the activation of downstream AKT targets (Fig. EV3A,B). Among them, a ~40 kDa protein retained our attention because its phosphorylation levels increased in cells transfected with all three variants and decreased upon treatment with alpelisib. Given its molecular weight, we hypothesized that this protein was most likely PRAS40, a canonical effector of AKT. We confirmed that the phosphorylation levels of PRAS40 were significantly increased in cells transfected with *PIK3R1* variants compared to wild-type *PIK3R1*, and returned to control levels upon treatment with alpelisib (Fig. 3A,B). However, we did not observe any changes in the phosphorylation levels of mTORC1 targets p70S6K and 4eBP1 (Fig. 3A,B) regardless of the variant transfected, which further supports the absence of activation of the mTORC1 pathway.

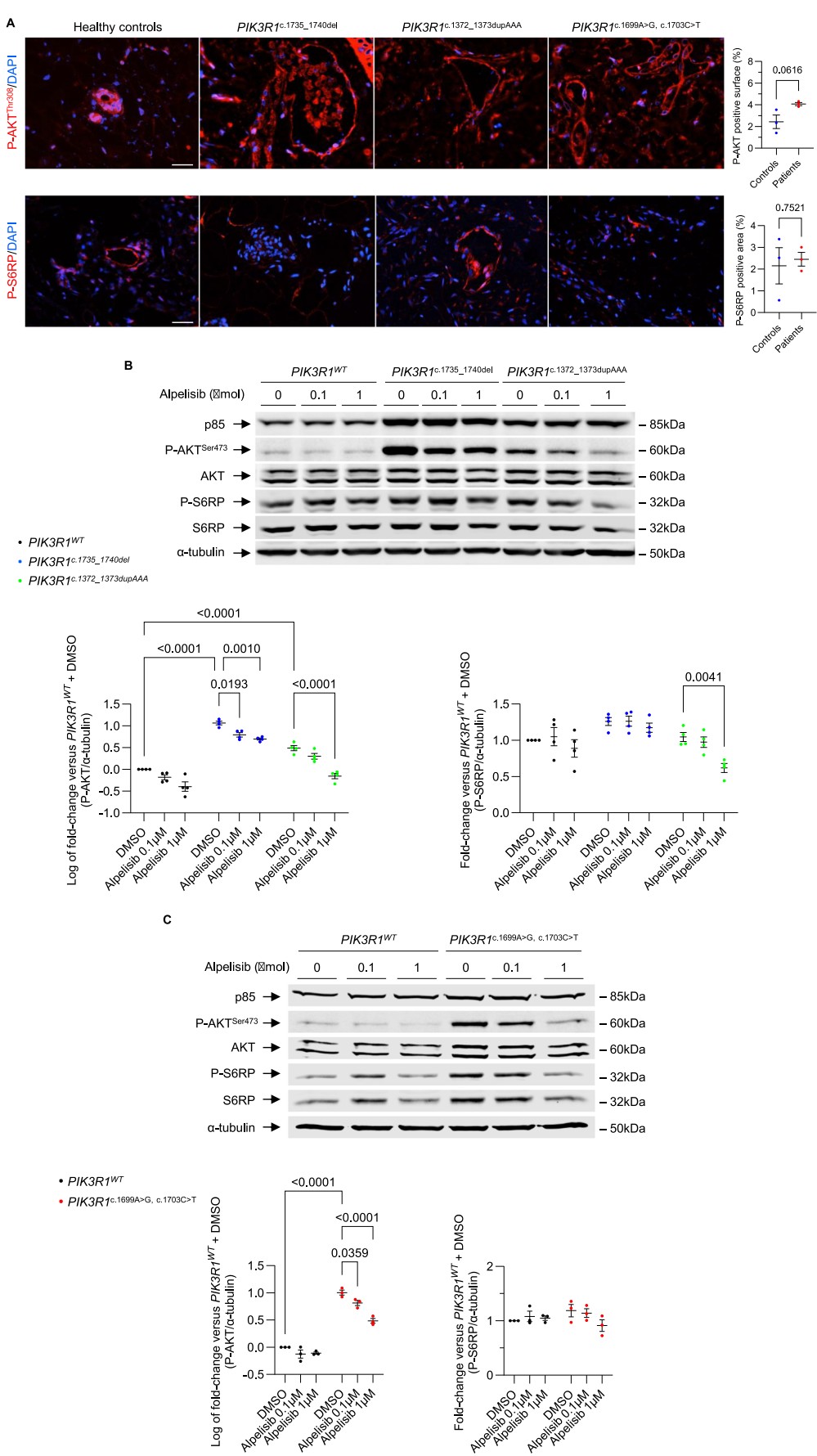

**Figure 1. PIK3R1 variants activate AKT and are sensitive to alpelisib.**

(A) Representative immunostainings of P-AKT$^{T308}$ and P-S6RP in skin vascular malformations from three patients and skin biopsies from healthy controls, and quantification. Scale bar 50 μm. Graphs show mean ± SEM from 3 biological replicates in each group. P-values were obtained from Student's t-tests. (B) Western blot and quantification of P-AKT$^{S473}$ and P-S6RP levels in HeLa cells transfected with plasmids containing either PIK3R1$^{WT}$, PIK3R1$^{c.1735\_1740del}$ or PIK3R1$^{c.1372\_1373dupAAA}$ variants, treated with either vehicle or alpelisib. In all experiments, cells were stimulated with recombinant human IGF-1 (10 ng/mL) for 30'. Graphs show mean ± SEM from 4 biological replicates in each group. P-values were obtained from two-way ANOVAs; exact p-values for P-AKT/α-tubulin are: $p = 3.81 \times 10^{-12}$ when comparing PIK3R1$^{c.1735\_1740del}$ (DMSO) and PIK3R1$^{WT}$ (DMSO), $p = 1.95 \times 10^{-5}$ when comparing PIK3R1$^{c.1372\_1373dupAAA}$ (DMSO) and PIK3R1$^{WT}$ (DMSO) and $p = 1.82 \times 10^{-7}$ when comparing PIK3R1$^{c.1372\_1373dupAAA}$ (alpelisib 1 μM) and PIK3R1$^{c.1372\_1373dupAAA}$ (DMSO). (C) Western blot and quantification of P-AKT$^{S473}$ and P-S6RP levels in HeLa cells transfected with plasmids containing either PIK3R1$^{WT}$ or PIK3R1$^{c.1699A>G, c.1703C>T}$ variant, treated with either vehicle or alpelisib. In all experiments, cells were stimulated with recombinant human IGF-1 (10 ng/mL) for 30'. Graphs show mean ± SEM from 3 biological replicates in each group. P-values were obtained from two-way ANOVAs; exact p-values for P-AKT/α-tubulin are: $p = 8.11 \times 10^{-9}$ when comparing PIK3R1$^{c.1699A>G, c.1703C>T}$ (DMSO) and PIK3R1$^{WT}$ (DMSO), $p = 1.12 \times 10^{-5}$ when comparing PIK3R1$^{c.1699A>G, c.1703C>T}$ (alpelisib 1 μM) and PIK3R1$^{c.1699A>G, c.1703C>T}$ (DMSO). Source data are available online for this figure.

## Patients with *PIK3R1*-related disorders demonstrated clinical, biological, and radiological improvements with alpelisib

Following these promising results, the three patients were included in French compassionate use program from the Agence Nationale de Sécurité du Médicament et des Produits de Santé (ANSM) and the Managed Access Program (MAP) from Novartis.

Patient 1, a 9-year-old boy, presented with disseminated lymphatic malformations (right arm, right axilla, chest, abdomen, pelvis, and left leg), complex venous and capillary malformations, asymmetrical overgrowth of the left leg with length discrepancy and scoliosis (Fig. 4A). He had recurrent episodes of lymphatic inflammatory flares with oozing, sepsis, chronic pain (pain visual analog scale at 7/10 permanently and weekly peak at 9/10), fatigue (grade 2), repeated thrombotic events and chronic elevated D-Dimer levels (Figs. 4B,C and EV4A,B). He was on anticoagulant medication with warfarin sodium and antibiotic prophylaxis (cefaclor). At the age of 7, a treatment with rapamycin was initiated during 6 months without success and then stopped. A skin biopsy from an area of a vascular malformation revealed the presence of a pathogenic *PIK3R1* c.1735_1740del variant (VAF 18%). Since the disease was progressing, we started alpelisib at a dose of 50 mg daily. Rapidly, the patient demonstrated improvement. Fatigue and chronic pain disappeared (grade 1 and pain visual analog score 0/10, respectively, at 6 months post treatment initiation), lymphatic malformations shrunk, complex vascular malformations discolored and efficacy remained sustained over time (Fig. 4A–C). Whole body MRI showed that the volume of the vascular malformation decreased by 23.4% and 36% (from 1909.04 to 1305.71 cm³) 6 and 12 months after alpelisib introduction, respectively (Fig. 4D,E). Biologically, we noticed complete correction of the D-Dimer levels (Fig. 4F). No adverse events related to alpelisib were reported during follow-up. Antibiotic and anticoagulation were withdrawn 6 months after treatment initiation.

Patient 2 was a 29-year-old man with asymmetrical overgrowth of the right leg and both feet, bilateral macrodactyly with syndactyly, disseminated capillary malformations, scoliosis, facial asymmetry, epilepsy, lymphatic and venous malformations of both limbs with chronic oozing, skin ulceration and bleeding, pain (pain visual analog score 9/10 permanently) treated with opioids, fatigue (grade 3) and chronic elevated D-Dimer levels (Figs. 4B,C,F,G and EV4A,B). He experienced several episodes of sepsis with cellulitis. The patient was treated with rapamycin during one year without efficacy. A skin biopsy from a superficial vascular malformation revealed the presence of a *PIK3R1* c.1372_1373dupAAA variant (VAF 15%). Because the disease was progressing, we decided to treat him with alpelisib at a dose of 250 mg per day. Following drug introduction, we noticed a rapid and sustained improvement in pain (pain visual analog score 0/10, 6 months after treatment initiation), fatigue (grade 1, 6 months after treatment initiation), oozing and bleeding stopped, skin ulcerations healed, vascular malformations were clinically improved as well as asymmetrical overgrowth (Fig. 4B,C,G). During the treatment, he had no epileptic nor septic episodes. MRI showed a 16% reduction (3352.42 to 2816.10 cm³) in the volume of the vascular malformations 12 months after alpelisib introduction (Fig. 4E,H). Biologically, we observed an improvement in D-Dimer levels (Fig. 4F). Biological exams did not show any signs of drug-related toxicity. The patient did not report adverse events.

Patient 3 was a 9-year-old boy with asymmetrical overgrowth of the right leg, macrodactyly of the 2nd, 3rd, and 4th fingers of the right hand, the right part of the face, disseminated complex slow-flow vascular malformations, chronic oozing, fatigue (grade 3), intense pain treated chronically with opioids (pain visual analog score 8/10 permanently with daily peak at 10/10), recurrent lymphatic inflammatory flares and sepsis treated with steroids and antibiotics and chronically elevated D-Dimer levels (Figs. 4B,C,F,I and EV4A,B). The patient was treated with rapamycin since the age of 2 with modest degree of efficacy (minor improvement in oozing). A skin biopsy from the vascular malformation revealed the presence of two pathogenic in cis *PIK3R1* variants, c.1699A>G (VAF 8%) and c.1703C>T (VAF 8%). Because the disease was progressing, we decided to stop rapamycin treatment and to have him start alpelisib 1 month later at a dose of 50 mg per day. Rapidly, we noticed a general improvement in his condition. Indeed, pain decreased and he could stop all medications including opioids within 1 month (pain visual analog scale 0/10, 12 months after treatment initiation) (Fig. 4B). Chronic fatigue improved (grade 1, 6 months after treatment initiation) (Fig. 4C). Mixed vascular malformations were clinically improved in terms of color, size and surface regularity (Fig. 4I). During the treatment period, he had no inflammatory flares, sepsis nor thrombosis. MRI revealed a 16% decrease in the volume of the vascular malformations (333.5 to 280.14 cm³) 12 months after alpelisib introduction (Fig. 4E,J). Biological exams showed D-Dimer levels correction (Fig. 4F) and no sign of drug-related toxicity. Neither the patient nor his family reported any adverse events related to alpelisib.

Overall, in these 3 patients, treatment with alpelisib was associated with a mean 22.67% (SD, 11.55) reduction in the

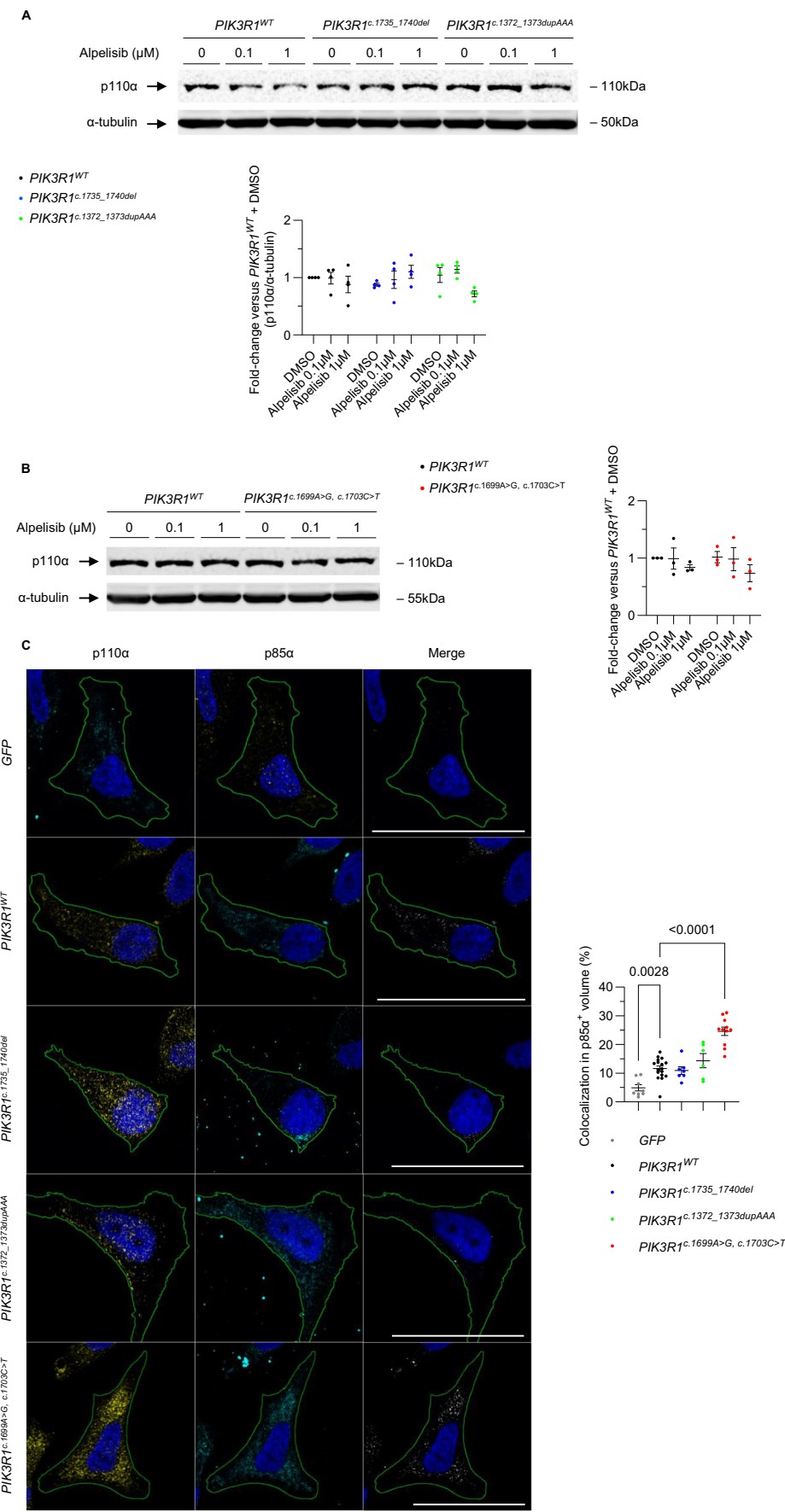

**Figure 2.** **p110α colocalizes with p85α in HeLa cells transfected with *PIK3R1* variants.**

(A) Western Blot and quantification of p110α levels in HeLa cells transfected with plasmids encoding *PIK3R1^WT^*, *PIK3R1^c.173S_1740del^* or *PIK3R1^c.1372_1373dupAAA^*. Graphs show mean ± SEM from 4 biological replicates in each group. *P*-values were obtained from two-way ANOVAs. (B) Western blot and quantification of p110α levels in HeLa cells transfected with plasmids encoding *PIK3R1^WT^* or *PIK3R1^c.1699A>G, c.1703C>T^*. Graphs show mean ± SEM from 3 biological replicates in each group. *P*-values were obtained from two-way ANOVAs. (C) Colocalization studies of p110α and p85α. Representative slice images of p110α, p85α and their colocalization mask in HeLa cells transfected with plasmids containing either *PIK3R1^WT^*, *PIK3R1^c.173S_1740del^*, *PIK3R1^c.1372_1373dupAAA^*, or *PIK3R1^c.1699A>G, c.1703C>T^* variants. Graph shows mean ± SEM from, respectively, 8, 17, 11, 7, or 6 biological replicates in each group. *P*-values were obtained from a one-way ANOVA: the exact *p*-value obtained when comparing *PIK3R1^c.1699A>G, c.1703C>T^* and *PIK3R1^WT^* is $p = 1.89 \times 10^{-9}$. Source data are available online for this figure.

volume of the vascular malformations as assessed by MRI after 12 months on alpelisib.

## Discussion

Here, we report the clinical, biological and radiological improvement in 3 patients with *PIK3R1*-related disorders treated with alpelisib, a PI3Kα inhibitor. We show that *PIK3R1* mutations in the iSH2 domain respond to PI3Kα inhibition in vitro and in patients. The iSH2 domain plays a crucial role in binding to the ABD domain of p110α, which inhibits the catalytic activity of p110α. Our results suggest that p85α mutants are still able to bind to p110α. Thus, the mutations reported may disrupt the iSH2-ABD interaction thereby relieving the inhibitory effect on p110α. Interestingly, *PIK3R1* mutations are commonly observed in solid cancer, highlighting the potential therapeutic implications of our findings in a broader context.

The degree of improvement observed in these 3 patients was comparable to what we have previously observed in patients with PROS (Morin and Canaud, 2021; Venot et al, 2018). It is noteworthy that these patients had previously undergone unsuccessful treatment with rapamycin. The lack of response to rapamycin could be attributed to the absence of mTORC1 activation by the specific *PIK3R1* variants we observed in vitro. This suggests that alternative downstream targets of AKT may be recruited due to *PIK3R1* mutations.

Following the EPIK P1 clinical trial (NCT04285723), alpelisib received accelerated approval from the US FDA in 2022 for PROS patients from the age of 2 (Canaud et al, 2023b). Given the absence of pharmacokinetic studies in PROS, the recommended alpelisib dosage is based on real-world experience. Alpelisib is associated with several adverse events including glucose metabolism disorders, alopecia, diarrhea, and vomiting. The EPIK P1 study reported less severe and less frequent adverse events, likely because of the lower dose used compared to cancer trials (Andre et al, 2019). Common adverse events in the EPIK P1 trial included alopecia, diarrhea, hyperglycemia, and aphthous ulcers (Canaud et al, 2023a).

This study has several limitations. Indeed, we do not have proper volumetric evaluation of the malformations using MRI during the rapamycin treatment period for the three patients to assess disease progression on this medication. Furthermore, the low number of patients in this case series requires to be cautious before drawing any conclusion. Finally, the variation in terms of volumetric response to alpelisib will deserve further investigations including pharmacokinetic studies.

In conclusion, this study represents a significant advancement in the field of personalized medicine. While further studies are needed

to validate these promising results, our findings pave the way for new therapeutic approaches in patients with overgrowth syndromes and vascular anomalies caused by *PIK3R1* variants involving the iSH2 domain.

## Methods

**Reagents and tools table**

| Reagent/Resource | Reference or Source | Identifier or Catalog Number |
| --- | --- | --- |
| **Experimental models** | | |
| HeLa cell line human | Merck | 93021013 |
| **Plasmids** | | |
| GFP | VectorBuilder | VB210708-1118hgr |
| hPIK3R1 WT + GFP | VectorBuilder | VB220330-1182cfm |
| hPIK3R1 p.(Q579-Y580)del + GFP | VectorBuilder | VB220330-1186cmc |
| hPIK3R1 p.K459dup + GFP | VectorBuilder | VB220330-1180ueb |
| hPIK3R1 WT | VectorBuilder | VB211115-1274xvk |
| hPIK3R1 p.K567E | VectorBuilder | VB211115-1275qgv |
| hPIK3R1 p.568L | VectorBuilder | VB211125-1031wsr |
| hPIK3R1 p.(K567E,P568L) | VectorBuilder | VB211125-1034sup |
| **Antibodies** | | |
| PI3 Kinase p85 Antibody | Cell Signaling | CS4292 |
| PI3 Kinase p110α (C73F8) Rabbit mAb | Cell Signaling | CS4249 |
| Phospho-Akt (Ser473) (D9E) XP® Rabbit mAb | Cell Signaling | CS4060 |
| Akt (pan) (40D4) Mouse mAb | Cell Signaling | CS2920 |
| Phospho-S6 Ribosomal Protein (Ser240/244) (D68F8) XP® Rabbit mAb | Cell Signaling | CS5364 |
| S6 Ribosomal Protein (54D2) Mouse mAb | Cell Signaling | CS2317 |
| Phospho-PRAS40 (Thr246) Antibody | Cell Signaling | CS2640 |
| Phospho-p70 S6 Kinase (Thr389) (108D2) Rabbit mAb | Cell Signaling | CS9234 |

| Reagent/Resource | Reference or Source | Identifier or Catalog Number |
|---|---|---|
| Phospho-4E-BP1 (Thr37/46) (236B4) Rabbit mAb | Cell Signaling | CS2855 |
| Anti-α-Smooth Muscle Actin (ACTA2) Antibody | Sigma-Aldrich | a5228 |
| Anti-TUBA4A (TUBA1) Antibody | Sigma-Aldrich | t5168 |
| Anti-Mouse StarBright Blue 700 Goat | Bio-Rad | 12004158 |
| Anti-Rabbit StarBright Blue 520 Goat | Bio-Rad | 12005869 |
| Anti-Rabbit IgG, HRP-linked Antibody | Cell Signaling | CS7074 |
| Phospho-Akt (Thr308) (C31E5E) Rabbit mAb | Cell Signaling | CS2965 |
| Alexa Fluor 555–conjugated secondary antibody | Thermo Fisher Scientific | a21206 |
| PI 3-kinase p85α Antibody (B-9) | Santa Cruz Biotechnology | sc-1637 |
| PI3KCA Polyclonal Antibody | Bioss Antibodies | BS2067R |
| Horse anti-mouse IgG biotinylated Antibody | Vector Laboratories | BA-2001 |
| Donkey anti-Rabbit IgG Highly Cross-Adsorbed Secondary Antibody Alexa Fluor 647 | Invitrogen | A31573 |
| Streptavidin Alexa Fluor 555 Conjugate | Invitrogen | S32355 |
| **Chemicals, Enzymes and other reagents** | | |
| Maxwell RSC DNA FFPE Kit | Promega France | AS1450 |
| SureSelectXT HS Custom Pannel | Agilent Technologies | |
| Fetal Bovine Serum | Sigma | F7524 |
| Penicillin-streptomycin | Gibco | 15140122 |
| DMEM | Gibco | 11965092 |
| Lipofectamine 3000 Reagent | Invitrogen | L3000008 |
| Alpelisib | MedChemExpress | HY-15244 |
| Recombinant human IGF-1 | Novus Biologicals | P05019 |
| RIPA buffer | Merck Millipore | 20-188 |
| cOmplete™ Protease Inhibitor Cocktail | Roche Diagnostics | 11697498001 |
| Phosphatase Inhibitor Cocktail 1 | Sigma-Aldrich | P2850 |
| 4X Laemmli buffer | Biorad | 1610747 |
| Beta-mercaptoethanol | Thermo Fisher Scientific | 35602BID |
| Transblot Turbo nitrocellulose membrane | Biorad | 1704271 |
| Bovine Serum Albumine | Euromedex | 1035-70-C |
| Tris-buffered saline-Tween 0.1% | Euromedex | ET220-B |
| Clarity Max Western ECL substrate | Biorad | 1705062 |
| Citrate buffer | Scytek | CBB999 |
| Tris-EDTA | Diagnostic BioSystems | K043 |
| Fluoromount | Sigma | F4680 |
| 4% paraformaldehyde (PFA) solution | Electron Microscopy Sciences | 15714S |

| Reagent/Resource | Reference or Source | Identifier or Catalog Number |
|---|---|---|
| Avidin/Biotin Blocking Kit | Golden Bridge International | E08-100 |
| DAPI | Invitrogen | D3571 |
| Microscope slides | Erie Scientific | 2951 |
| **Software** | | |
| Image Lab software | Bio-Rad | version 6.0.1 |
| ImageJ | ImageJ | version 1.54f |
| Ilastik software | Ilastik | version 1.4.0.post1 |
| Fiji software | Fiji | version 1.54f |
| AlphaFold Server | Google DeepMind | version 3 |
| PyMOL software | Schrödinger | version 2.5.7 |
| 3D Slicer software | 3D Slicer | version 5.6.2 |
| Prism software | GraphPad | version 10.2.3 |
| **Other** | | |
| MiSeq platform | Illumina | |
| ChemiDoc MP imager | Bio-Rad | 12003154 |
| Eclipse Ni-E microscope | Nikon | Ni-E |
| SP8-STED confocal microscope | Leica | TCS SP8 STED |

## Methods and protocols

### Genetic sequencing

DNA was extracted from formalin-fixed, paraffin-embedded (FFPE) or frozen tissue samples with the Maxwell RSC DNA FFPE Kit (Promega France) and was analyzed by targeted Next-Generation Sequencing (NGS) on a panel of 23 genes involved in vascular malformations and overgrowth syndromes (*AKT1*, *AKT2*, *AKT3*, *BRAF*, *EPHB4*, *GNA11*, *GNA14*, *GNAQ*, *HRAS*, *KRAS*, *KRIT1*, *MAP2K1*, *MAP3K3*, *MTOR*, *NRAS*, *PIK3CA*, *PIK3R1*, *PIK3R2*, *PTEN*, *RASA1*, *TEK*, *TSC1*, and *TSC2*). Briefly, all coding exons of selected genes were screened by paired-end sequencing reactions of 150-bp reads on a MiSeq platform (Illumina, Paris, France) after capture-based target enrichment (Agilent Technologies). Bioinformatic analyses included trimming of raw NGS reads (FASTQ), mapping, variant calling, variant annotation, and filtering. Somatic variant curation was performed as recommended in cancer (Horak et al, 2022). The variant allele frequency threshold was set at 1%.

### Plasmids

Plasmids encoding WT and variants p.(K567E,P568L), p.(Q579-Y580)del, and p.K459dup of the human *PIK3R1* gene were constructed by VectorBuilder upon request of the authors. All plasmid structures and references (see Reagents and Tools) are described in Table EV1 and available on https://vectorbuilder.com.

### Cell culture

HeLa cells (passage 9 to 26, Merck, ref. 93021013, tested negative for mycoplasma contamination) were cultured in 10% fetal bovine

**A**

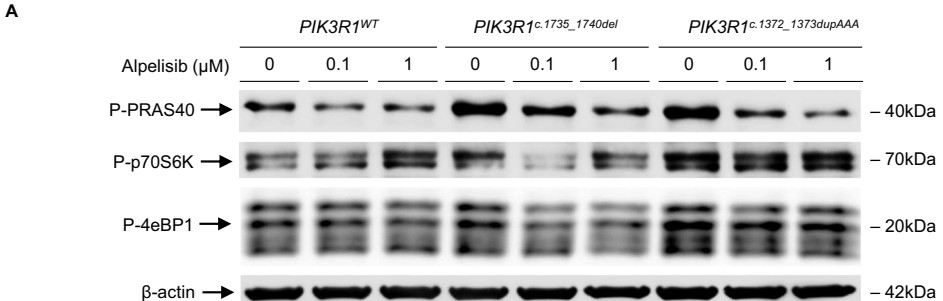

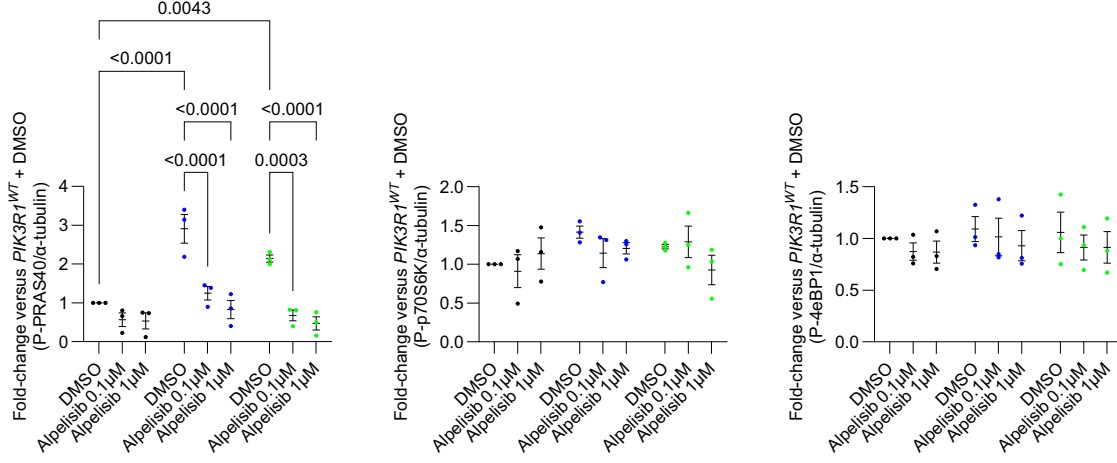

**B**

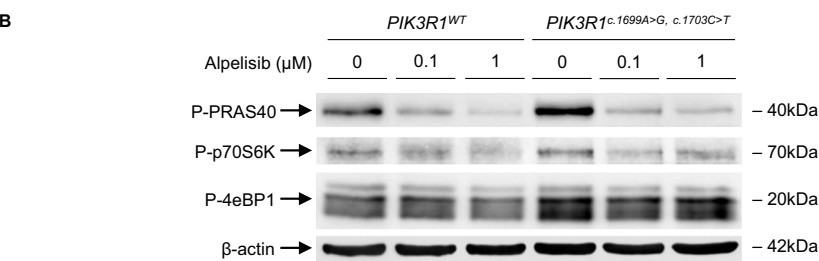

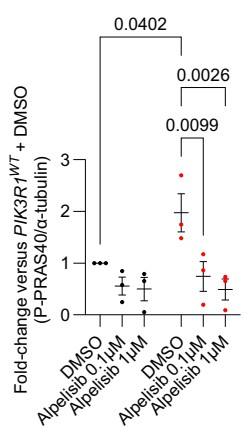

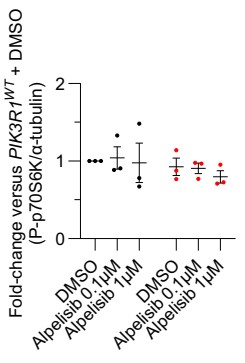

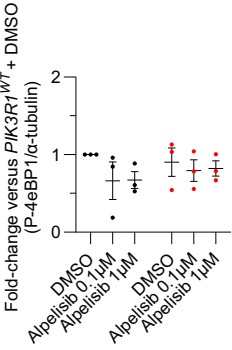

◄ **Figure 3. PRAS40, but not mTORC1 effectors, is phosphorylated in HeLa cells transfected with *PIK3R1* variants and responds to alpelisib.**

(A) Western blot and quantification of phosphorylation levels of AKT target PRAS40, and mTORC1 targets p70S6K and 4eBP1 in HeLa cells transfected with plasmids coding for either *PIK3R1*<sup>WT</sup>, *PIK3R1*<sup>c.1735_1740del</sup> or *PIK3R1*<sup>c.1372_1373dupAAA</sup> variants. Graphs show mean ± SEM from 3 biological replicates in each group. *P*-values were obtained from two-way ANOVAs; exact *p*-values for P-PRAS40/α-tubulin are: $p = 1.34 \times 10^{-5}$ when comparing *PIK3R1*<sup>c.1735_1740del</sup> (DMSO) and *PIK3R1*<sup>WT</sup> (DMSO), $p = 7.89 \times 10^{-5}$ when comparing *PIK3R1*<sup>c.1735_1740del</sup> (alpelisib 0.1 μM) and *PIK3R1*<sup>c.1735_1740del</sup> (DMSO), $p = 4.22 \times 10^{-6}$ when comparing *PIK3R1*<sup>c.1735_1740del</sup> (alpelisib 1 μM) and *PIK3R1*<sup>c.1735_1740del</sup> (DMSO) and $p = 7.37 \times 10^{-5}$ when comparing *PIK3R1*<sup>c.1372_1373dupAAA</sup> (alpelisib 1 μM) and *PIK3R1*<sup>c.1372_1373dupAAA</sup> (DMSO). (B) Western blot and quantification of phosphorylation levels of PRAS40, p70S6K and 4eBP1 in HeLa cells transfected with plasmids coding for either *PIK3R1*<sup>WT</sup> or *PIK3R1*<sup>c.1699A>G, c.1703C>T</sup> variant. Graphs show mean ± SEM from 3 biological replicates in each group. *P*-values were obtained from two-way ANOVAs. Source data are available online for this figure.

serum (FBS; ref. F7524) and 1% penicillin-streptomycin (Gibco, ref. 15140122) supplemented DMEM (Gibco, ref. 11965092) and seeded in 6-well plates (0.25 million per well) 24 h before transfection. Briefly, cells were transfected with 0.5 μg of plasmid DNA delivered with 1 μL of Lipofectamine 3000 Reagent (Invitrogen, ref. L3000008) per well in 10% FBS supplemented, antibiotic-free DMEM. 24 h later, the transfection medium was replaced with fresh FBS-supplemented DMEM. Transfected cells were then starved for 15 h, treated for 1 h with alpelisib (0.1 or 1 μM, MedChemExpress, ref. HY-15244) or an equivalent volume of DMSO, and stimulated with 10 ng/mL recombinant human IGF-1 (Novus Biologicals, ref. P05019) over the last 30 min of treatment. Cells were then immediately washed with PBS and lysed with RIPA buffer (Merck Millipore, ref. 20-188) containing protease and phosphatase inhibitors (Roche Diagnostics, ref. 11697498001; Sigma, ref. P2850).

### Western blotting

Cell lysates were incubated at 4 °C on a rotating wheel for 30 min, then centrifuged at 14,000 rcf for 15 min at 4 °C, and supernatant was collected. Protein denaturation was obtained by dilution in 4X Laemmli buffer (Bio-Rad, ref. 1610747) with beta-mercaptoethanol (Thermo Fisher Scientific, ref. 35602BID) and heating at 95 °C for 5 min. For each sample, 20 μg of proteins were resolved on a 10% polyacrylamide gel and transferred onto a nitrocellulose membrane (Transblot Turbo, Bio-Rad, ref. 1704271). The membranes were cut in order to isolate molecular weights of interest, blocked with 3% bovine serum albumin (BSA; Euromedex, ref. 1035-70-C) in Tris-buffered saline-Tween 0.1% (TBS-T, Euromedex, ref. ET220-B), and incubated with primary antibodies directed against p85α (Cell Signaling, ref. CS4292), p110α (Cell Signaling, ref. CS4249), phospho-AKT<sup>S473</sup> (Cell Signaling, ref. CS4060), pan-AKT (Cell Signaling, ref. CS2920), phospho-S6RP (Cell Signaling, ref. CS5364), S6RP (Cell Signaling, ref. CS2317), phospho-PRAS40 (Cell Signaling, ref. CS2640), phospho-p70S6K (Cell Signaling, ref. CS9234), phospho-4eBP1 (Cell Signaling, ref. CS2855), actin (Sigma, ref. a5228), or α-tubulin (Sigma, ref. t5168) at 4 °C overnight. All primary antibodies were diluted 1:1000 in blocking buffer, except for actin and α-tubulin, which were diluted at 1:10,000. The day after, the membranes were washed with TBS-T, then incubated for 45 min with secondary antibodies (anti-Mouse StarBright Blue 700 Goat, Bio-Rad, ref. 12004158; Anti-Rabbit StarBright Blue 520 Goat, Bio-Rad, ref. 12005869; or anti-rabbit HRP-linked antibody, Cell Signaling, ref. CS7074) at a 1:10,000 dilution in TBS-T, then washed again with TBS-T. The membranes were finally incubated with Clarity Max Western ECL substrate (ref. 1705062, Bio-Rad) for 5 min when HRP-linked secondary antibodies were used. Protein bands were then photographed with a

ChemiDoc MP imager (Bio-Rad, ref. 12003154) and analyzed with the Image Lab software (version 6.0.1, Bio-Rad).

### Immunofluorescence

FFPE tissue sections (4 μm) were deparaffinized and submitted to antigen retrieval protocols at high temperature (120 °C) and high pressure in a pressure cooker in citrate buffer (Scytek, ref. CBB999) for P-S6RP staining, or Tris-EDTA buffer (Diagnostic BioSystems, ref. K043) for P-AKT staining. Sections were then blocked with 3% BSA and 10% FBS in TBS-T and incubated with primary antibodies (1:100 rabbit anti-P-AKT<sup>T308</sup>, Cell Signaling, ref. CS2965; 1:50 rabbit anti-P-S6RP<sup>S240/244</sup>, Cell Signaling, ref. CS5364). After washing with TBS-T, the slides were incubated with Alexa Fluor 555–555-conjugated secondary antibody (Thermo Fisher Scientific, ref a21206) diluted 1:200 in blocking buffer, counterstained with DAPI, washed again in TBS-T before mounting with Fluoromount (Sigma, ref. F4680) and imaged using an Eclipse Ni-E microscope (Nikon).

P-S6RP and P-AKT<sup>T308</sup> signal in human skin samples were analyzed with ImageJ (v1.54f). After background subtraction, an automatic threshold was applied to each image to discriminate positive and negative areas (https://imagej.net/plugins/auto-threshold#Moments). To obtain a better assessment of AKT and S6RP phosphorylation in blood vessels and soft tissues, areas containing skin structures interfering with staining analysis (epidermis and hair follicles) were excluded by hand-contouring. P-AKT and P-S6RP-positive surface areas were assessed by dividing the positive surface by the total surface analyzed in each image and expressed as percentages. The figure pictures were selected among representative images of the statistical results. As control, we used 3 biopsies from 3 healthy donors.

### Colocalization analysis

HeLa cells (passage 9 to 26) were plated on coverslips and transfected as previously described with 2 μg of plasmid DNA encoding wild-type or mutant *PIK3R1*. The culture medium was changed 24 h later, then the cells were starved for 15 h with 1% penicillin-streptomycin-supplemented DMEM without FBS. 48 h after transfection, cells were washed with PBS and fixed in 4% paraformaldehyde (PFA; Electron Microscopy Sciences, ref. 15714S) for 10 min.

Afterwards, each coverslip was washed with phosphate buffered saline (PBS), before avidin and biotin blocking (Avidin/Biotin Blocking Kit, Golden Bridge International, ref. E08-100) followed by blocking with BSA (3% in TBS-T) and 10% FBS for 30 min. The coverslips were then incubated overnight at 4 °C with primary antibodies directed against p85α (Santa Cruz Biotechnology, ref. sc-1637) and p110α (Bioss, ref. BS2067R) diluted 1:50 and 1:200 in blocking buffer, respectively. Afterwards, the coverslips were

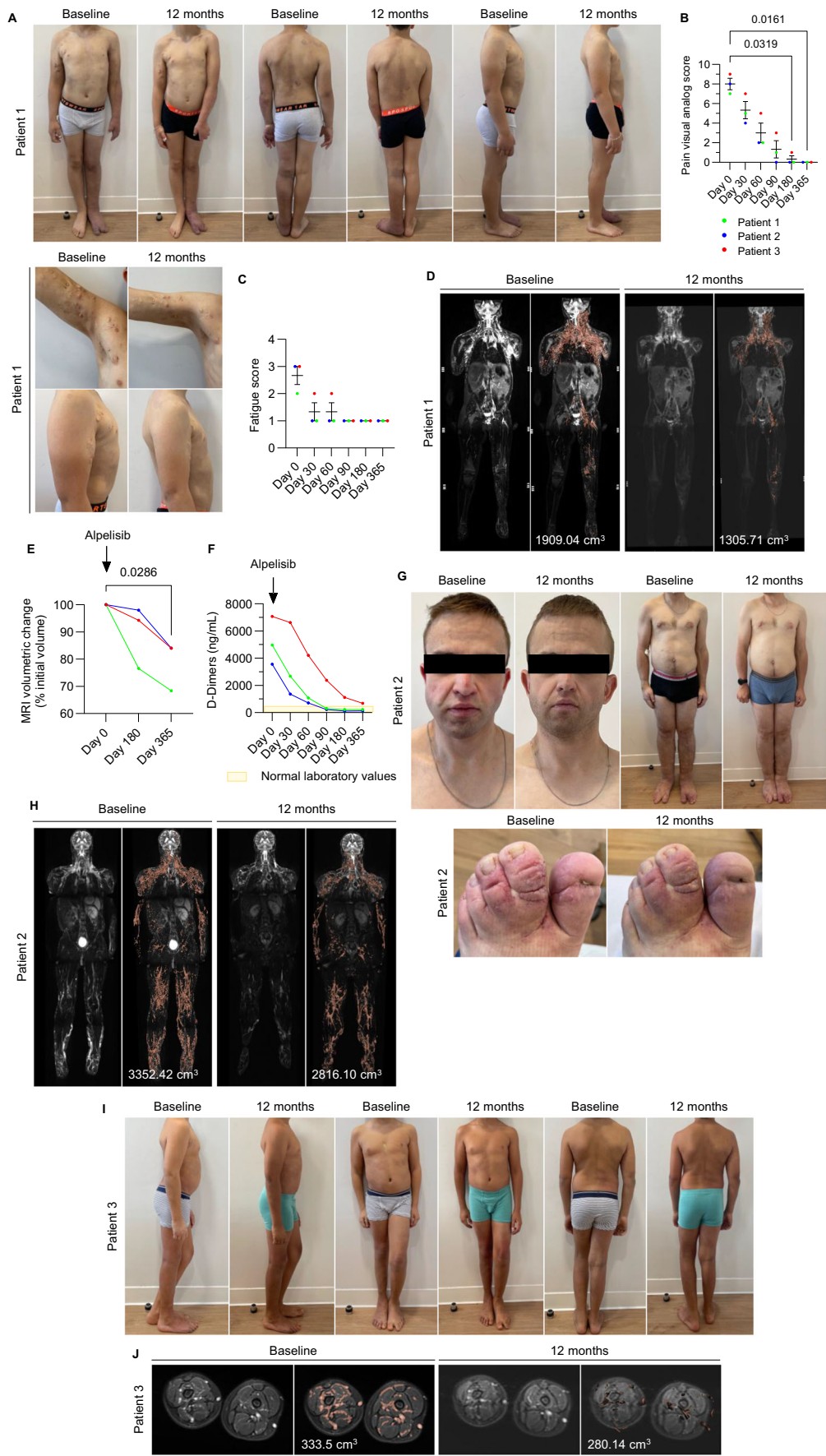

**Figure 4. Alpelisib improves the condition of patients with *PIK3R1* variants.**

(A) Representative pictures of the morphological changes observed in patient 1 receiving alpelisib for 12 months. (B) Pain visual analog score. Graph shows mean ± SEM from 3 patients. *P*-values were obtained from a Friedman test. (C) Fatigue score. Graph shows mean ± SEM from 3 patients. No statistical difference was seen between timepoints according to a Friedman test. (D) Coronal T2 with fat saturation MRI and 3D segmentation of vascular malformations observed in patient 1 before and after alpelisib introduction. (E) Percentage change of the volume of vascular malformations in the 3 patients. P-values were obtained from a Friedman test. (F) D-Dimer levels before and following alpelisib introduction. Graph shows longitudinal data from 3 patients and normal laboratory values. (G) Representative pictures of the morphological changes observed in patient 2 receiving alpelisib for 12 months. (H) Coronal T2 with fat saturation MRI and 3D segmentation of vascular malformations observed in patient 2 before and after alpelisib introduction. (I) Representative pictures of the morphological changes observed in patient 3 receiving alpelisib for 12 months. (J) Axial T2 with fat saturation MRI and 3D segmentation of vascular malformations observed in patient 3 before and after alpelisib introduction. Source data are available online for this figure.

washed with PBS-T, incubated for 45 min with a horse anti-mouse IgG biotinylated antibody (Vector Laboratories, ref. BA-2001) diluted at 1:500 in blocking buffer at room temperature, washed again with PBS-T, and finally incubated with fluorescent secondary antibodies (Donkey anti-Rabbit IgG Highly Cross-Adsorbed Secondary Antibody Alexa Fluor 647, Invitrogen, ref. A31573; Streptavidin Alexa Fluor 555 Conjugate, Invitrogen, ref. S32355) diluted at 1:500 and 1:200 in blocking buffer, respectively. After washing with PBS-T, cells were stained with DAPI (Invitrogen, ref. D3571) for 5 min, and washed again before mounting with Fluoromount Aqueous Mounting Medium (Sigma, ref. F4680) on microscope slides (Erie Scientific, ref. 2951). Slides were and kept in a dark and dry place, at 4 °C until imaging z-stacks (300 nm spacing) with a SP8-STED confocal microscope (Leica). Pixel classification was performed by machine learning (Ilastik software, version 1.4.0.post1) after training on 2–3 images per condition to identify p85α and p110α positive signals. Segmentation masks were generated for p85α and p110α signals from pixel classification using Fiji software (version 1.54f), and cells were manually contoured to restrict analyzed regions. Colocalization surfaces of both proteins were calculated as the intersecting area of both masks on each slice. The colocalization volume was obtained by z-summation of colocalization surfaces and expressed relative to p85α-positive volume in each contoured cell.

### Protein modeling

Protein sequences were obtained from UniProt (https://www.uniprot.org/) for both wild-type p110α (UniProt: P42336; Ref Seq: NP_006209.2) and wild-type p85α (UniProt: A0A2X0SFG1). The variants sequences were obtained by manually modifying the amino acid sequence according to each patient's mutation. Structures predictions of the PI3Kα complex containing wild-type p110α and either wild-type or mutant p85α were then generated using AlphaFold3 server (Abramson et al, 2024). The output includes a heatmap that shows the predicted aligned error for each position relative to another and assesses the reliability of the model. The root mean square deviation (RMSD) of atomic positions was calculated using PyMOL software (Schrödinger, version 2.5.7) to compare three-dimensional protein foldings. Each variant model was superimposed to the wild-type model and the RMSD of the entire dimer was calculated using the command line *align wt_protein, variant_protein*. Dimers conformations were considered similar if their RMSD values were <1 Å.

### Patients

This study was conducted on 3 patients, including 2 children who were followed at *Hôpital Necker Enfants Malades*, following procedures previously described elsewhere (Delestre et al, 2021). All patients with vascular malformations and/or overgrowth syndrome and a somatic *PIK3R1* variant receiving alpelisib in our center were included. Treatment administration was performed according to the protocol approved by the ANSM and the Novartis Managed Access Program. Written informed consent was obtained from adult patients and from the parents of pediatric patients. Alpelisib was compassionately offered by Novartis. Adult patients received 250 mg/day, and pediatric patients received 50 mg/day (Venot et al, 2018). Alpelisib was taken orally every morning during breakfast. Patients were assessed at regular intervals as previously reported (Venot et al, 2018). Adverse events were graded according to Common Terminology Criteria for Adverse Events [CTCAE], version 4.03, and coded by preferred term using the Medical Dictionary for Regulatory Activities [MedDRA], version 24.0. Biological exams were performed monthly during the first 3 months and then every 3 months. They included complete blood count, D-Dimers, hemostasis parameters, blood electrolytes, kidney and liver function, glycated hemoglobin, cholesterol, and triglycerides measurements. All experiments conducted in patients conformed to the principles set out in the WMA Declaration of Helsinki and the Department of Health and Human Services Belmont Report. Consent to publish photographs was obtained from all three patients.

### Imaging

Patients underwent full body magnetic resonance imaging (MRI) prior to alpelisib introduction, and this exam was repeated 6 and 12 months after drug introduction. MRI examination was performed using T1, T2, T2 with fat suppression weighted imaging sequences. Vascular volumetric evaluation on MRI was performed with 3D Slicer software with manual segmentation tools (Fedorov et al, 2012). Volumes were calculated by summing images based on 2D contours and slice thickness.

### Statistical analysis

For each *PIK3R1* variant, at least three different independent cell culture experiments were used for statistical analysis. All cell experiments were included. Student's t-tests were used to compare continuous variables in two groups. One-way ANOVAs were used to compare three of more groups. When testing the effect of two factors (i.e., *PIK3R1* variants and treatment conditions), two-way ANOVAs were used. When using ANOVAs, normal distribution of residues was assessed using the D'Agostino-Pearson omnibus ($K^2$) test, or the Shapiro-Wilk's test when the sample size was insufficient. Homoscedasticity of residues was verified using the Brown-Forsythe test for one-way ANOVA, and the Spearman's test for heteroscedasticity for two-way ANOVA. In case of violation of homoscedasticity, values were log-transformed before analysis (P-AKT levels, Fig. 1B,C).

## The paper explained

### Problem

Recently, somatic *PIK3R1* mutations have been discovered in patients with overgrowth syndromes and complex vascular malformations. *PIK3R1* encodes for p85a, a protein that participates in the control of cell growth and proliferation through the PI3Kα/AKT/mTOR pathway. Care of patients with *PIK3R1*-related disorders mainly relies on symptomatic treatments.

### Results

Here, we identified three patients with somatic *PIK3R1* mutations. We conducted in vitro experiments to examine the effects of these mutations and found that p85α mutants were sensitive to alpelisib, a PI3Kα inhibitor. Based on these results, we treated the three patients with alpelisib. Following drug initiation, all patients demonstrated clinical and biological improvement associated with a significant reduction in malformation volume as assessed by MRI. The drug was well tolerated in the 3 patients.

### Impact

This study supports PI3Kα inhibition as a promising therapeutic approach for individuals with *PIK3R1*-related anomalies.

When assessing efficacy of alpelisib in patients, no blinding nor randomization were applied in the absence of a control group, and sample size was not calculated. Consecutive measurements of bioclinical parameters (pain visual analog score (VAS), fatigue VAS, D-Dimers, platelets, fibrinogen; Figs. 4C–F and EV4A,B) were compared using the Friedman test. A *p*-value below 0.05 was considered significant and Šidák's correction was applied to relevant multiple pairwise comparisons. All statistical analyses were performed using the GraphPad Prism software (version 10.2.3).

## Data availability

This study includes no data deposited in external repositories.

The source data of this paper are collected in the following database record: biostudies:S-SCDT-10_1038-S44321-025-00249-9.

## Peer review information

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

## Acknowledgements

This study was supported by the European Research Council (CoG 2020 grant number 101000948 awarded to GC), the Agence Nationale de la Recherche – Programme d'Investissements d'Avenir (ANR-18-RHUS-005 to GC) and the Agence Nationale de la Recherche – Programme de Recherche Collaborative (19-CE14-0030-01 to GC). This work was also supported by the CLOVES SYNDROME COMMUNITY (West Kennebunk, USA), Association Syndrome de CLOVES (Nantes, France), Fondation d'entreprise IRCEM (Roubaix, France), Fonds de dotation Emmanuel BOUSSARD (Paris, France), MSD Avenir - grant Signalopathies (Paris, France), the Fondation DAY SOLVAY (Paris, France), the Fondation TOURRE (Paris, France) to GC, the Fondation BETTENCOURT SCHUELLER (Paris, France) to GC, the Fondation Simone et Cino DEL DUCA (Paris, France), the Fondation Line RENAUD-Loulou GASTE (Paris, France), Fondation Schlumberger pour l'Education et la Recherche (Paris, France), Fondation Maladies Rares, the Association Robert Debré pour la Recherche Médicale awarded to GC, WonderFIL smiles - a Facial Infiltrating Lipomatosis community (Norway), INSERM, Assistance Publique Hôpitaux de Paris, Université Paris Cité, Fondation pour la Recherche Médicale (FDM202006011222) awarded to GM and Banting Postdoctoral Fellowship (Canadian Institutes of Health Research, #472149) awarded to APG. We are also very grateful to our generous donors. Alpelisib was supplied by Novartis at no cost as a part of the Managed Access Program.

## Author contributions

**Gabriel Morin**: Data curation; Formal analysis; Validation; Investigation; Visualization; Methodology; Writing—original draft; Writing—review and editing. **Alexandre P Garneau**: Data curation; Formal analysis; Validation; Investigation; Visualization; Methodology; Writing—original draft; Writing—review and editing. **Nabiha Bouzakher**: Data curation; Formal analysis; Investigation. **Louise Ségot**: Data curation; Software; Formal analysis; Validation; Investigation; Methodology. **Antoine Fraissenon**: Data curation; Formal analysis; Investigation. **Amélie Blondel**: Data curation; Formal analysis; Investigation. **Florence Petit**: Data curation; Formal analysis; Investigation. **Caroline Chopinet**: Data curation; Investigation. **Franck Mayeux**: Data curation; Software; Investigation. **Pierre Fayoux**: Data curation; Investigation. **Anne Dompmartin**: Investigation. **Christine Bodemer**: Investigation. **Estelle Balducci**: Data curation; Supervision; Validation; Methodology. **Sophie Kaltenbach**: Investigation. **Patrick Villarese**: Data curation; Formal analysis; Investigation. **Vahid Asnafi**: Data curation; Formal analysis; Investigation. **Christophe Legendre**: Data curation; Investigation. **Christine Broissand**: Data curation; Investigation. **Sylvie Fraitag**: Data curation; Formal analysis; Investigation. **Chloé Quelin**: Data curation; Investigation. **Nicolas Goudin**: Software; Formal analysis; Validation; Visualization. **Laurent Guibaud**: Data curation; Investigation. **Guillaume Canaud**: Conceptualization; Data curation; Formal analysis; Supervision; Funding acquisition; Validation; Investigation; Visualization; Methodology; Writing—original draft; Project administration; Writing—review and editing.

Source data underlying figure panels in this paper may have individual authorship assigned. Where available, figure panel/source data authorship is listed in the following database record: biostudies:S-SCDT-10_1038-S44321-025-00249-9.

## Disclosure and competing interests statement

A patent application ("BYL719 (alpelisib) for use in the treatment of PIK3CA-related overgrowth spectrum" #WO2017140828A1) has been filed by INSERM (Institut National de la Santé et de la Recherche Médicale), Centre National De La Recherche Scientifique (CNRS), Université Paris Cité, and Assistance Publique-Hôpitaux De Paris (AP-HP) for the use of BYL719 (alpelisib) in the treatment of *PIK3CA*-related overgrowth spectrum (PROS/CLOVES syndrome). G. Canaud is the inventor. This patent is licensed to Novartis. Guillaume Canaud receives or has received consulting fees from Novartis, Fresenius Medical Care, Vaderis, Alkermes, IPSEN and BridgeBio. The other authors declare no other competing interests. Guillaume Canaud is an editorial advisory board member. This has no bearing on the editorial consideration of this article for publication.

# Expanded View Figures

**Figure EV1. Double mutant in cis is associated with a strong stimulation of P-AKT compared to single mutants.**

Western blot and quantification of AKT phosphorylation on residue Ser$^{473}$ and S6RP in HeLa cells transfected with plasmids containing either GFP, *PIK3R1*$^{WT}$, *PIK3R1*$^{c.1699A>G}$, *PIK3R1* $^{c.1703C>T}$ or *PIK3R1*$^{c.1699A>G, c.1703C>T}$ variants. In all experiments, cells were stimulated with recombinant human IGF-1 (10 ng/mL) for 30'. Graphs show mean ± SEM from 3 biological replicates in each group. *P*-values were obtained from two-way ANOVAs; exact *p*-values for P-AKT$^{S473}$/tubulin are $p = 9.35 \times 10^{-5}$ when comparing *PIK3R1*$^{c.1699A>G, c.1703C>T}$ and *PIK3R1*$^{WT}$; $p = 7.30 \times 10^{-5}$ when comparing *PIK3R1*$^{c.1699A>G, c.1703C>T}$ and *PIK3R1*$^{c.1703C>T}$. Exact *p*-values for P-AKT$^{S473}$/total AKT are $p = 1.17 \times 10^{-5}$ when comparing *PIK3R1*$^{c.1699A>G, c.1703C>T}$ and *PIK3R1*$^{WT}$; $p = 1.09 \times 10^{-5}$ when comparing *PIK3R1*$^{c.1699A>G, c.1703C>T}$ and *PIK3R1*$^{c.1703C>T}$ Source data are available online for this figure.

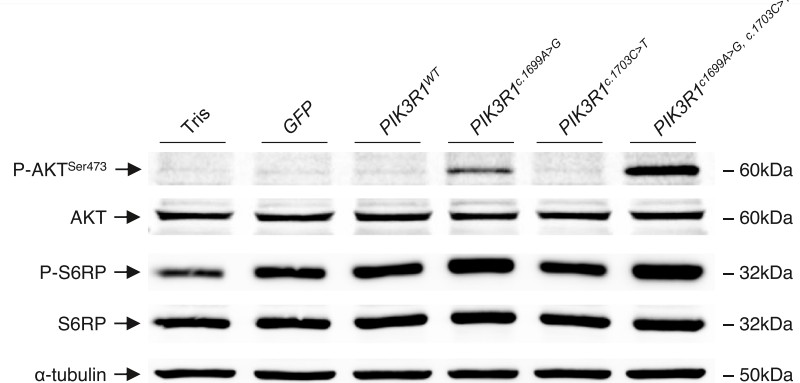

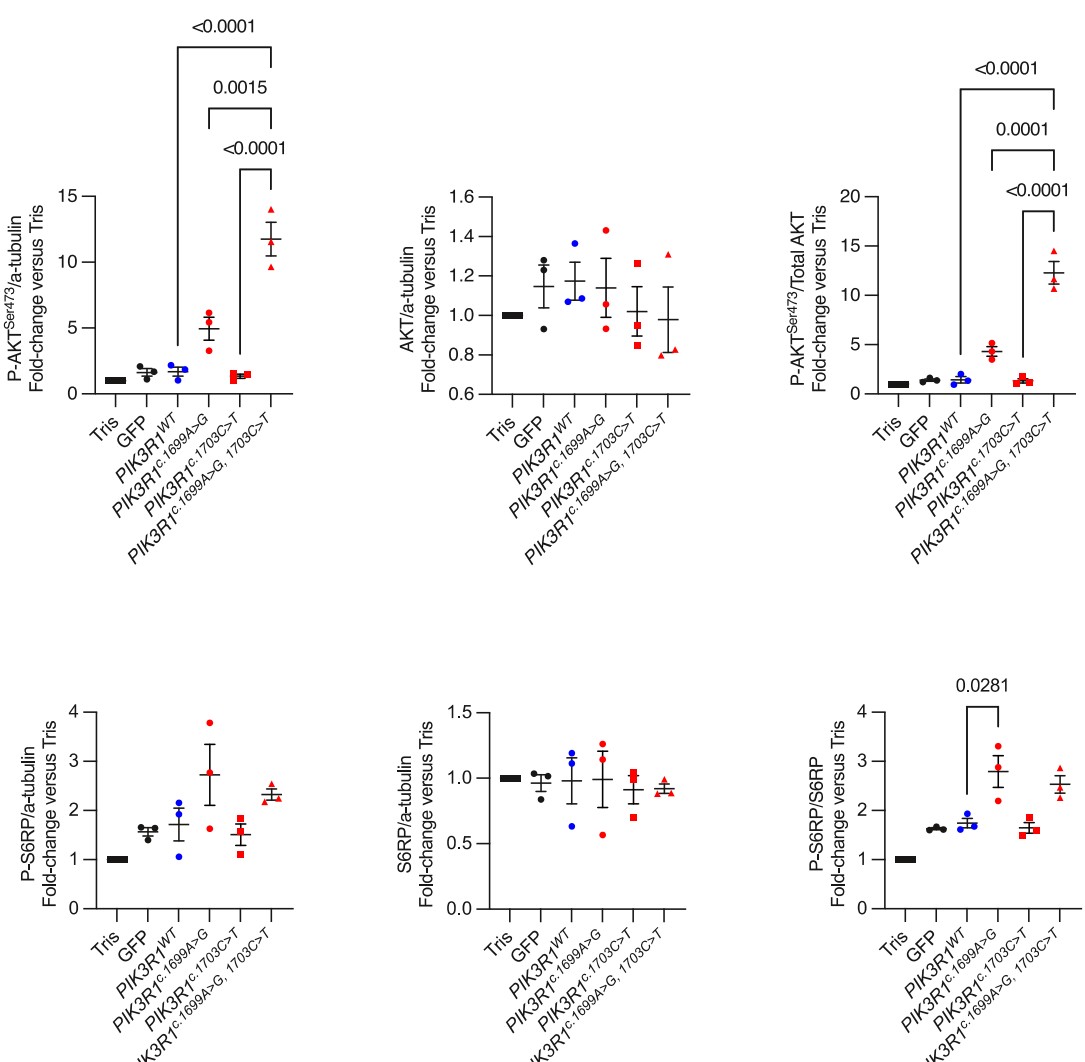

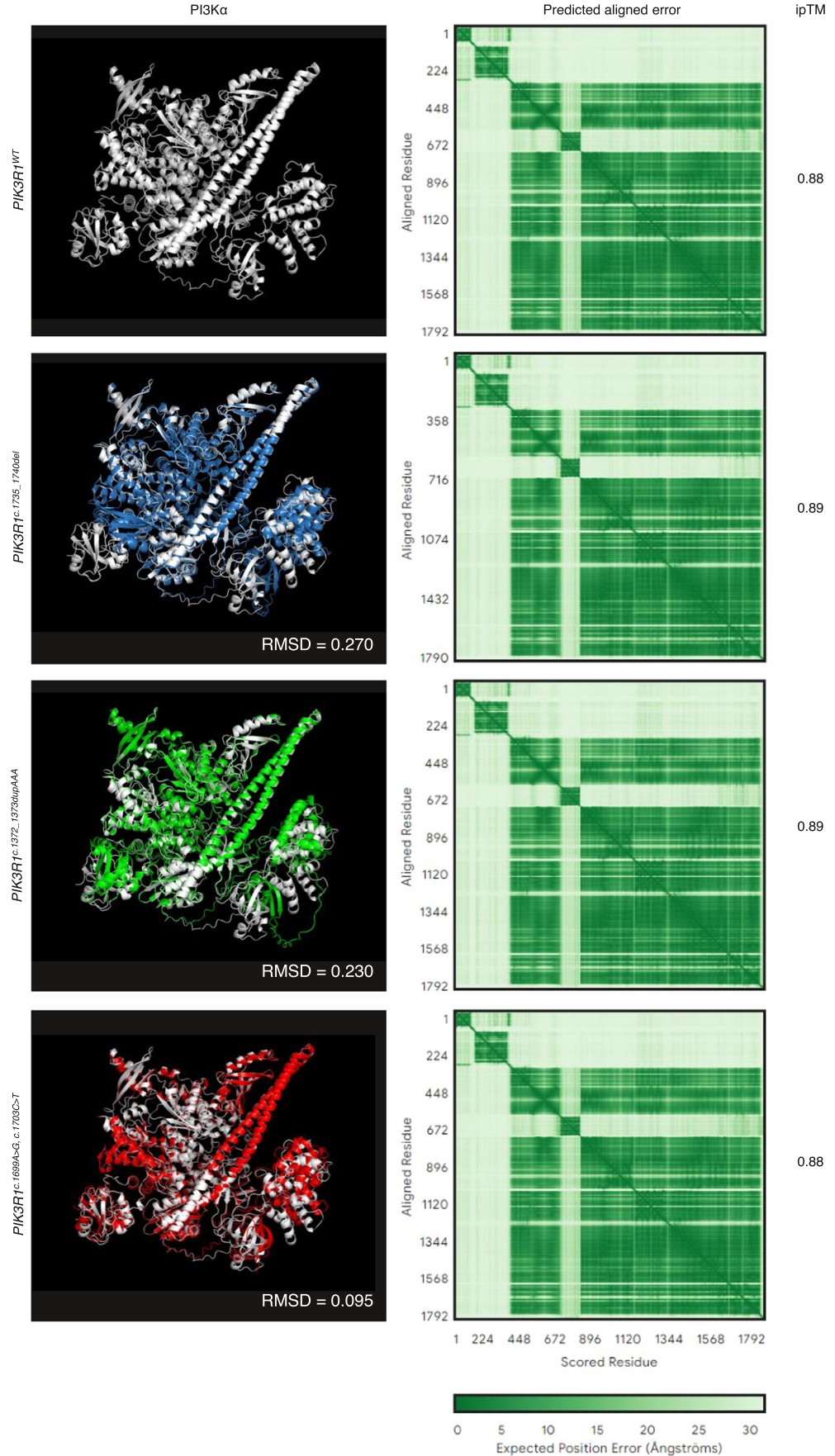

◀ **Figure EV2. 3D modeling of PI3Kα resulting from the dimerization of wild-type p110α with the different p85α mutants.**

Left panel: 3D structures of PI3Kα containing wild-type p110α and either wild-type p85α or variants (*c.1735_1740del, c.1372_1373dupAAA* or *c.1699A>G, c.1703C>T*), as predicted by AlphaFold3 and formatted with the PyMOL software. The resulting 3D model of each variant was superimposed to that of wild-type PI3Kα for comparison. Root mean square deviation (RMSD) values are shown for each mutant compared to the wild-type dimer. Right panel: Predicted aligned error graphs obtained from AlphaFold3 show high reliability of the predicted models regarding the position of the iSH2 domain of p85α (approx. residues 448 to 672) relative to the p110α subunit (approx. residues 896 to 1792). Interface predicted template modeling (ipTM) scores show highly reliable positioning of the two subunits in the predicted dimers. Source data are available online for this figure.

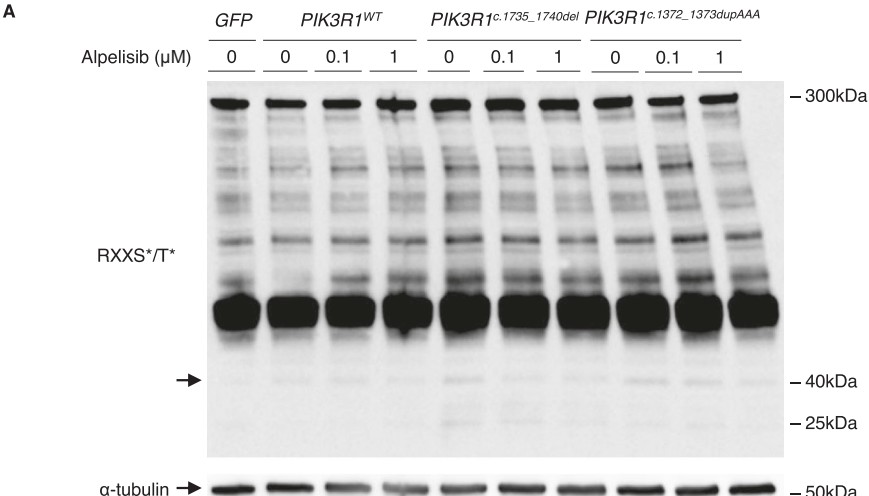

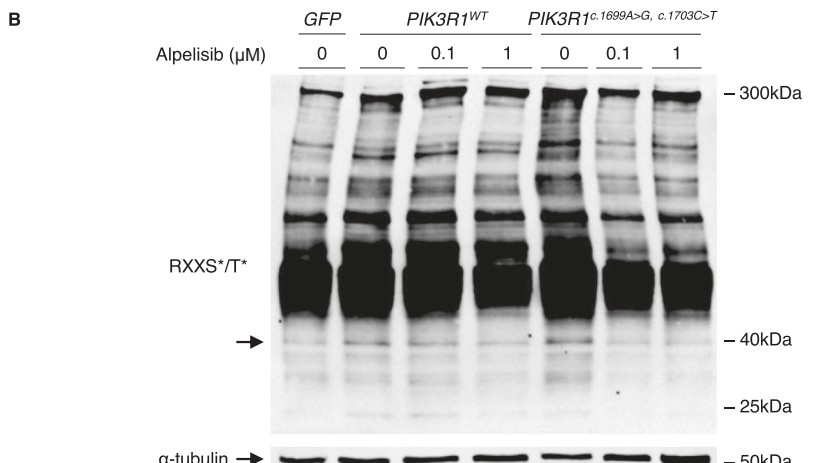

**Figure EV3. Phosphorylation profiles of AKT targets identify a 40 kDa effector sensitive to alpelisib in cells transfected with *PIK3R1* variants.**

(A) Western Blot of proteins phosphorylated on RXXS*/T* residues in HeLa cells transfected with plasmids encoding *PIK3R1^WT^*, *PIK3R1^c.1735_1740del^* or *PIK3R1^c.1372_1373dupAAA^*. (B) Western Blot of proteins phosphorylated on RXXS*/T* residues in HeLa cells transfected with plasmids encoding *PIK3R1^WT^* or *PIK3R1^c.1699A>G, c.1703C>T^*. In both cases, the phosphorylation levels of a 40 kDa protein increase in cells transfected with *PIK3R1* variants and decrease upon treatment with alpelisib. Source data are available online for this figure.

A

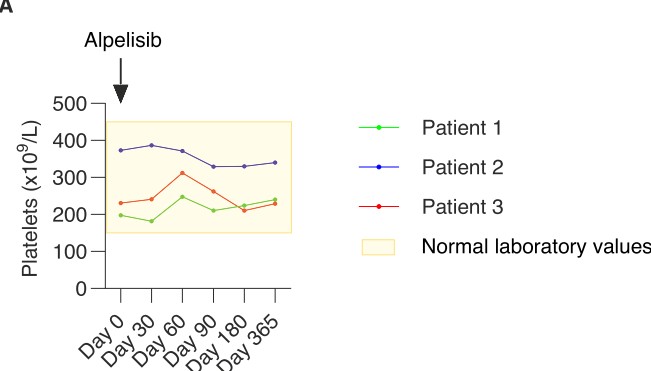

B

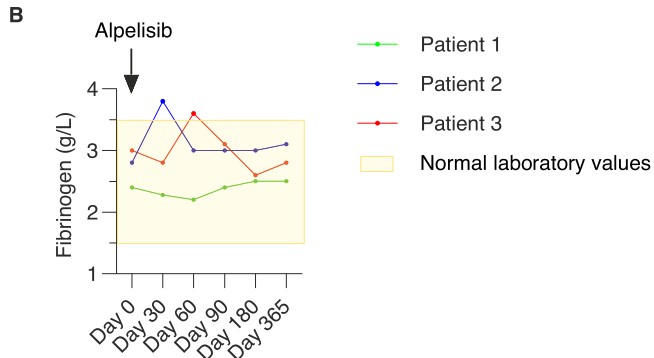

**Figure EV4.  Patients had normal platelets counts and fibrinogen throughout alpelisib treatment.**

(**A**) Platelet counts before and following alpelisib introduction. Graph shows longitudinal data from 3 patients and normal laboratory values. No statistical difference was seen between timepoints according to a Friedman test. (**B**) Fibrinogen levels before and following alpelisib introduction. Graph shows longitudinal data from 3 patients and normal laboratory values. No statistical difference was seen between timepoints according to a Friedman test. Source data are available online for this figure.

