## [Peer Review File · EMBO Molecular Medicine]

Somatic PIK3R1 mutations in the iSH2 domain are accessible to PIK3CA inhibition

Gabriel Morin, Alexandre Garneau, Nabiha Bouzakher, Louise Segot, Antoine Fraissenon, Amelie Blondel, Florence Petit, Caroline Chopinet, Franck Mayeux, Pierre Fayoux, Anne Domp martin, Christine Bodemer, Estelle Balducci, Sophie Kaltenbach, Patrick Villarese, Vahid Asnafi, Christophe Legendre, Christine Broissand, Sylvie Fraitag, Chloe QUELIN, nicolas goudin, Laurent Guibaud, and Guillaume Canaud

Corresponding author: Guillaume Canaud (guillaume.canaud@inserm.fr)

Review Timeline:

Submission Date:	3rd Oct 23
Editorial Decision:	27th Oct 23
Revision Received:	14th Jun 24
Editorial Decision:	17th Jul 24
Revision Received:	3rd Apr 25
Accepted:	24th Apr 25

Editor: Lise Roth

Transaction Report:

27th Oct 2023

Dear Dr. Canaud,

Thank you for submitting your work to EMBO Molecular Medicine. We have now heard back from the referees who agreed to evaluate your manuscript. As you will see below, the reviewers find that the question addressed by the study is of clinical interest, however they remain unconvinced that some of the major conclusions are sufficiently supported by the data. After further discussion, the referees agreed that the following points should be addressed in a major revision of the manuscript:

- deeper characterization of the lack of mTORC1 recruitment in a more relevant cellular model
- more detailed information on the clinical aspects, including the methods of genetic testing

If you feel you can satisfactorily address these points and those listed by the referees, you may wish to submit a revised version of your manuscript. Please attach a covering letter giving details of the way in which you have handled each of the points raised by the referees. A revised manuscript will once again be subject to review, and we cannot guarantee at this stage that the eventual outcome will be favorable.

We are expecting your revised manuscript within three months, if you anticipate any delay, please contact us.

We require:

4) A .docx formatted letter INCLUDING the reviewers' reports and your detailed point-by-point responses to their comments. As part of the EMBO Press transparent editorial process, the point-by-point response is part of the Review Process File (RPF), which will be published alongside your paper.

5) A complete author checklist, which you can download from our author guidelines (<https://www.embopress.org/page/journal/17574684/authorguide#submissionofrevisions>). Please insert information in the checklist that is also reflected in the manuscript. The completed author checklist will also be part of the RPF.

6) Please note that all corresponding authors are required to supply an ORCID ID for their name upon submission of a revised manuscript.

7) It is mandatory to include a 'Data Availability' section after the Materials and Methods. Before submitting your revision, primary datasets produced in this study need to be deposited in an appropriate public database, and the accession numbers and database listed under 'Data Availability'. Please remember to provide a reviewer password if the datasets are not yet public (see <https://www.embopress.org/page/journal/17574684/authorguide#dataavailability>).

8) For data quantification: please specify the name of the statistical test used to generate error bars and P values, the number (n) of independent experiments (specify technical or biological replicates) underlying each data point and the test used to calculate p-values in each figure legend. The figure legends should contain a basic description of n, P and the test applied. Graphs must include a description of the bars and the error bars (s.d., s.e.m.). Please provide exact p values.

9) Our journal encourages inclusion of *data citations in the reference list* to directly cite datasets that were re-used and

obtained from public databases. Data citations in the article text are distinct from normal bibliographical citations and should directly link to the database records from which the data can be accessed. In the main text, data citations are formatted as follows: "Data ref: Smith et al, 2001" or "Data ref: NCBI Sequence Read Archive PRJNA342805, 2017". In the Reference list, data citations must be labeled with "[DATASET]". A data reference must provide the database name, accession number/identifiers and a resolvable link to the landing page from which the data can be accessed at the end of the reference. Further instructions are available at .

13) Author contributions: CRediT has replaced the traditional author contributions section because it offers a systematic machine readable author contributions format that allows for more effective research assessment. Please remove the Authors Contributions from the manuscript and use the free text boxes beneath each contributing author's name in our system to add specific details on the author's contribution. More information is available in our guide to authors.

16) As part of the EMBO Publications transparent editorial process initiative (see our Editorial at <http://embomolmed.embopress.org/content/2/9/329>), EMBO Molecular Medicine will publish online a Review Process File (RPF) to accompany accepted manuscripts.

In the event of acceptance, this file will be published in conjunction with your paper and will include the anonymous referee reports, your point-by-point response and all pertinent correspondence relating to the manuscript. Let us know whether you agree with the publication of the RPF and as here, if you want to remove or not any figures from it prior to publication.

I look forward to receiving your revised manuscript.

Yours sincerely,

Lise Roth

***** Reviewer's comments *****

Referee #1 (Comments on Novelty/Model System for Author):

In vitro systems are not adequate.

Referee #1 (Remarks for Author):

In this study, Morin et al provide support for the use of Alpelisib for patients with overgrowth and vascular anomalies carrying somatic mutations in PIK3R1. While this is an interesting study of personalized medicine, it feels more like a clinical case report of PIK3R1 mutations in overgrowth syndromes with vascular anomalies. From a biological and mechanistic perspective, this study is poorly developed and contains some inaccuracies (even interpretation data from biopsies).

I am very worried about some of the overstatements done in the manuscript, especially in the context of mTOR. The data/conclusions presented may lead clinicians to think that rapamycin is not good for these patients; and as such stop trying the use of rapamycin for these patients. I think this would be a tremendous mistake, especially considering that the overstatements in this context have been done on the basis of not very reliable signaling studies including the use of a very artificial model (HELA + overexpression of the mutations). It would be good to validate the signaling data in patient-derived cells.

Have the authors discarded that these patients do not carry an activating mutation on PIK3CA? It is not clear for the data provided which is the impact of these mutations on PI3Ka? Neither on the implications of p85a. Are the levels of p85a changed by the mutation? Is p85a still bound to p110a? Which is the impact of such mutations in p110a levels?

How did authors make the quantification of pAkt and pS6 in Figure 1? Taking into account the whole tissue section or specifically to blood vessels? From the images provided it does not seem that pAkt levels and pS6 intensity is higher in the malformed vessels compared to controls. Cd31 staining is not clear in Fig. 1B.

Along the manuscript, there are often sentences which are not straight forward to understand: i.e. What does "recruitment" mean in the sentence (By immunofluorescence, we observed that AKT recruitment was recruited (line 17, page 10)? Same, what do author mean with "It is important to highlight that none of these variants exhibited mTORC1 recruitment" (line 8 in page11).

Authors should consider referring to original research articles rather than reviews along the manuscript.

Referee #2 (Comments on Novelty/Model System for Author):

This is an important paper for the extending indication of alpelisib in lateralized overgrowth syndrome.

Referee #2 (Remarks for Author):

This is an important paper to extend the indication of alpelisib in lateralized overgrowth syndrome. However, there are some points to be revised in the current version.

1. The authors did not mention about the safety profiles of each patients during alpelisib treatment in Results and Discussion. Especially adult patient might have diabetes.
The adverse events and management should be documented in Results and discussed in Discussion.
2. The variants should be classified by ACMG/AMP guidelines.
3. The information about the methods to detect the somatic variants and in which genes were included for analysis is missing and needs to be documented.
4. The flat dosing (50 mg qd) in the pediatric patients appears not relevant considering wide spectrum of body weight or BSA among the pediatric population.
The authors' opinion on this would be valuable for the real-world experience.
5. The information is missing how the double mutations were confirmed in cis position in the plasmid before the transfection in Hela Cells.

Referee #3 (Comments on Novelty/Model System for Author):

The model used is appropriate for the present functional studies. However, a greater characterization to validate the lack of mTORC1 recruitment would be a useful addition.

Referee #3 (Remarks for Author):

The authors describe three cases of PIK3R1-related overgrowth and PI3Kalpha inhibition with alpelisib in both the clinic and in vitro. This report adds both the number of cases of PIK3R1-related overgrowth in the literature and to the previous anecdote of alpelisib in PIK3R1 disorders. The preclinical assays provide interesting observations of an mTORC1 independent pathology that is possibly translated in the lack of sirolimus efficacy in the clinic.

Pg4 Line 11 - "The first two domains bind to the phosphatase and tensin homologue deleted on chromosome 10". What is meant by "deleted on chromosome 10"?

Pg10 Line 18 - The data presented include a lack of pS6RP but not mTORC1 directly, thus, absence of mTORC1 is inferred by lack of pS6RP. This should be made clearer in the manuscript. Did the authors consider evaluating other elements of the AKT/mTOR signaling such as PRAS40.

Pg11 Line 20 - Were other markers DIC such as fibrinogen, coagulopathy, thrombocytopenia abnormal at baseline? If so, did they also change during treatment with alpelisib?

Pg12 Line 8 - "The patient did not report adverse events." Adverse events should not only be the patient report but the laboratory monitoring (e.g., hyperglycemia) and clinician observations (e.g., rash). It would be helpful to specifically include what type of monitoring patients received. In addition, it is unlikely that there were no adverse events in three individuals over a 12-month period. It may, however, be possible that no drug-related adverse events occurred. Please clarify how adverse events are evaluated and reported.

Pg 12 - Are the authors able to comment on the trajectory of pain and fatigue improvement? For example, over the first 6 months, was their rapid enduring improvement in these symptoms or gradual linear improvement. Were formal assessments of pain and symptoms only at 6 and 12 months? The authors use the word 'rapid' multiple times in the manuscript but do not provide data on short interval follow up to support this except for the 1-month opioid follow up in patient 3.

Pg13 Line 9 - How is pain assessed? Was the accompanied question using the VAS for pain asked at time of clinic visit? Greatest pain in past 24 hours? Greatest pain in past week? Or were multiple questions asked? There is discrepancy in patient 3's pain being specified as a 'daily peak' but patient 1 as a weekly peak, and in patient 2 no peak was provided?

Pg14 Line 3 - Given only three points of data pre and three post, providing an interquartile range is not particularly informative. A simple list in the text or a table would be more appropriate.

Pg15 - I would recommend the authors discuss more about the limitations of this study. This may include discussion on the volumetric analysis, lack of data during rapamycin treatment (MRI volumetrics are not presented during sirolimus treatment), and other considerations.

Figure 1 A - CD31 is listed in the figure but is not included as a primary antibody in the methods section. Additionally, there does

not appear to be any CD31 positive cells in the figure. Do the authors have an explanation for why the few pS6RP positive cells are not also CD31 positive?

Referee #1 (Comments on Novelty/Model System for Author): In vitro systems are not adequate.

Referee #1 (Remarks for Author):

In this study, Morin et al provide support for the use of Alpelisib for patients with overgrowth and vascular anomalies carrying somatic mutations in PIK3R1. While this is an interesting study of personalize medicine, it feels more like a clinical case report of PIK3R1 mutations in overgrowth syndromes with vascular anomalies. From a biological and mechanistic perspective, this study is poorly developed and contains some inaccuracies (even interpretation data from biopsies).

We would like to thank Referee #1 for finding this article an “*interesting study of personalized medicine*”. To clarify why so few patients were involved in this study, we have to recall that *PIK3R1*-related mosaic disorders are very rare disorders, even compared to *PIK3CA*-related disorders (PRDs). Consistently with other reports(Kuentz et al., 2024, Siegel et al., 2018, Cottrell et al., 2021), at our institution, these diseases are far less frequent than PRDs(Morin et al., 2024) (approximately 1:25 in our center, 2.9% versus 73% of positive genetic samples in patients with overgrowth and/or vascular malformations).

I am very worried about some of the overstatements done in the manuscript, especially in the context of mTOR. The data/conclusions presented may lead clinicians to think that rapamycin is not good for these patients; and as such stop trying the use of rapamycin for these patients. I think this would be a tremendous mistake, especially considering that the overstatements in this context have been done on the basis of not very reliable signaling studies including the use of a very artificial model (HELA + overexpression of the mutations). It would be good to validate the signaling data in patient-derived cells.

We express our gratitude to the Reviewer for his/her comments. However, we find it surprising to encounter this statement. Up to this point, there have been no published medical reports outlining the utilization of rapamycin in individuals afflicted with *PIK3R1*-related vascular malformations. Any use of this medication by physicians would constitute an off-label application, carrying inherent risks of serious adverse effects and immune system suppression. Currently, Alpelisib stands as the sole drug approved by the US FDA for patients with *PIK3CA*-Related Overgrowth Spectrum. While our manuscript did not explicitly advocate for the universal treatment of all patients with *PIK3R1*-related disorders with alpelisib, we maintain the perspective that this remains the most suitable course of action.

In order to further support our findings, we carried out additional experiments. To this end, we transfected HeLa cells with the different variants and investigated the AKT substrates using Western blot analysis. We observed that the 3 variants produced a similar pattern of phosphorylated proteins, indicating the activation of downstream AKT targets (**new Figure Expanded View 3A and 3B**). Among them, a ~40kDa protein retained our attention because its phosphorylation levels increased in cells transfected with all three variants and decreased upon treatment with alpelisib. Given its molecular weight, we hypothesized that this protein was most likely PRAS40, a canonical effector of AKT. We confirmed that the phosphorylation levels of PRAS40 were significantly increased in cells transfected with *PIK3R1* variants compared to wild-type *PIK3R1*, and returned to control levels upon treatment with alpelisib (**Figure**

Expanded View 3A and 3B). However, we did not observe any changes in the phosphorylation levels of mTORC1 targets p70S6K and 4eBP1 (**Figure Expanded View 3A and 3B**) regardless of the variant transfected, which further supports the absence of activation of the mTORC1 pathway.

Altogether, these findings support that mTOR inhibitors may not be the most appropriate drugs for patients with *PIK3R1*-related disorders. These new data have been added to the revised version of the manuscript.

Have the authors discarded that these patients do not carry an activating mutation on PIK3CA? It is not clear for the data provided which is the impact of these mutations on PI3Ka? Neither on the implications of p85a. Are the levels of p85a changed by the mutation? Is p85a still bound to p110a? Which is the impact of such mutations in p110a levels?

In our practice, we employ a panel encompassing 23 genes linked to overgrowth syndromes and vascular anomalies (inclusive list: AKT1, AKT2, AKT3, BRAF, EPHB4, GNA11, GNA14, GNAQ, HRAS, KRAS, KRIT1, MAP2K1, MAP3K3, MTOR, NRAS, PIK3CA, PIK3R1, PIK3R2, PTEN, RASA1, TEK, TSC1, TSC2). We systematically investigated *PIK3CA* variants across our patients and did not uncover any additional variants in these three specific cases.

As depicted in **Figures 1B-1C** and **Figure 2A-2B**, no discernible differences in expression levels were observed for p85a or P110a. Additionally, as per the request, we have unsuccessfully tried to characterize the interaction between the variant of p85a and p110a using immunoprecipitation. However, using AlphaFold3 and PyMOL softwares, we were able to predict protein structures of both p85 α and the resulting PI3K α heterodimers for all variants when associated with wild-type p110 α . We computed the RMSD (Root-Mean-Square Deviation) values of p85 α , of wild-type p110 α in each heterodimer, and of the whole heterodimer for each variant. We observed that the resulting PI3K α dimers showed very little difference (RMSD values from 0.09 to 0.27Å). These predictions do not reveal conformational changes that could result in a loss of affinity of p85 α for p110 α . By co-immunofluorescence studies, we further confirmed that p85 α colocalizes with p110 α at least as much in HeLa cells transfected with *PIK3R1* variants as in cells expressing the wild-type gene (**Figure 2C**).

These new data have been added to the revised version of the manuscript.

How did authors make the quantification of pAkt and pS6 in Figure 1? Taking into account the whole tissue section or specifically to blood vessels? From the images provided it does not seem that pAkt levels and pS6 intensity is higher in the malformed vessels compared to controls. Cd31 staining is not clear in Fig. 1B.

As detailed in the Method section, the quantification of immunofluorescence was performed as follows:

“P-S6RP and P-AKT^{T308} signal in human skin samples were analyzed with ImageJ (v1.54f). After background subtraction, an automatic threshold was applied to each image to discriminate positive and negative areas (<https://imagej.net/plugins/auto-threshold#Moments>). To obtain a better assessment of AKT and S6RP phosphorylation in blood vessels and soft tissues, areas containing skin structures interfering with staining analysis (epidermis and hair follicles) were excluded by hand-

contouring. P-AKT and P-S6RP-positive surface areas were assessed by dividing the positive surface by the total surface analyzed in each image and expressed as percentages. The figure pictures were selected among representative images of the statistical results. As control, we used 3 biopsies from 3 healthy volunteers."

Along the manuscript, there are often sentences which are not straight forward to understand: i.e. What does "recruitment" mean in the sentence (By immunofluorescence, we observed that AKT recruitment was recruited (line 17, page 10)? Same, what do author mean with "It is important to highlight that none of these variants exhibited mTORC1 recruitment" (line 8 in page11).

We apologize if this was unclear. We have now modified the word "recruitment" for "activation".

Authors should consider referring to original research articles rather than reviews along the manuscript.

As requested, we have updated the reference list.

Referee #2 (Comments on Novelty/Model System for Author): This is an important paper for the extending indication of alpelisib in lateralized overgrowth syndrome.

We would to thank the Reviewer for his/her comments.

Referee #2 (Remarks for Author):

This is an important paper to extend the indication of alpelisib in lateralized overgrowth syndrome. However, there are some points to be revised in the current version.

1. The authors did not mention about the safety profiles of each patients during alpelisib treatment in Results and Discussion. Especially adult patient might have diabetes.

The adverse events and management should be documented in Results and discussed in Discussion.

We appreciate the Reviewer highlighting this significant aspect. In our results section, we documented the adverse events experienced by each patient. As of now, none of the three patients have reported any side effects linked to the medication. Adverse events associated with Alpelisib are extensively documented thanks to oncology and PROS clinical trials. These encompass side effects such as hyperglycemia, diabetes, rash, as identified in the EPIK P1 and P3 (long-term assessment of patients from the EPIK P1 trial) clinical trials. We have now expanded this information in the discussion section.

“Thanks to real-world evidence from the EPIK P1 clinical trial (NCT04285723), alpelisib has recently received accelerated approval from the US FDA for patients with PROS (age >2 years)(Canaud et al., 2023b, Canaud et al., 2023a). Alpelisib is associated with adverse events such as hyperglycemia, alopecia, diarrhea, and vomiting. However, in the EPIK P1 clinical trial, adverse events were less severe and less frequent, likely because of the lower dose used clinical trials(Andre et al., 2019). The most common adverse events in the EPIK P1 trial were grade 1 alopecia (16.7%), diarrhea (15.8%), hyperglycemia (12.3%), and aphthous ulcers (10.5%)⁵. It is important to note that the doses of alpelisib used here are based on real-world data, as no pharmacokinetics analyses were available.”

2. The variants should be classified by ACMG/AMP guidelines.

As requested, in the revised version of the manuscript the variants are now classified following the ACMG/AMP guidelines.

“We identified 3 patients with somatic PIK3R1 mutations and severe associated disorders (see below for the complete cases description). Patient 1 had a PIK3R1 c.1735_1740del variant (VAF 18%, ACMG class 5, pathogenic) (p.[Q579-Y580]del), patient 2 had a PIK3R1 c.1372_1373dupAAA variant (VAF 15%, ACMG class 5, pathogenic) (p.K459dup) and patient 3 had a double PIK3R1 mutation including c.1699A>G (VAF 8%, ACMG class 5, pathogenic) and c.1703C>T variants (VAF 8% ACMG class 4, likely pathogenic) in cis (p.[K567E,P568L]).”

3. The information about the methods to detect the somatic variants and in which genes were included for analysis is missing and needs to be documented.

The Reviewer brings up a crucial point. In our practice, we employ NGS technology across all our patients, with a variant allele frequency threshold set at 1%. Our panel encompasses 23 genes associated with overgrowth syndromes and vascular anomalies, including the following genes: AKT1, AKT2, AKT3, BRAF, EPHB4, GNA11, GNA14, GNAQ, HRAS, KRAS, KRIT1, MAP2K1, MAP3K3, MTOR, NRAS, PIK3CA, PIK3R1, PIK3R2, PTEN, RASA1, TEK, TSC1, and TSC2.

This refined description is now incorporated into the revised version of the manuscript.

4. The flat dosing (50 mg qd) in the pediatric patients appears not relevant considering wide spectrum of body weight or BSA among the pediatric population. The authors' opinion on this would be valuable for the real-world experience.

We thank the Reviewer for highlighting this consideration. The dosage utilized in this study is derived from the EPIK P1 clinical trial conducted among patients with PIK3CA-related overgrowth spectrum. In EPIK P1, this lower dosage demonstrated notable clinical and radiological improvements along with an acceptable safety profile. As EPIK P1 constitutes a real-world data-based clinical trial, there is an absence of pharmacokinetic data within this population.

The Reviewer is correct in suggesting potential dose adaptation, considering the circumstances. However, given our observed clinical, biological, and radiological improvements without encountering drug-related adverse events, we opted to maintain this dosage for the two pediatric patients. It's important to note that despite reassuring data in the midterm follow-up of these patients, we remain exceedingly cautious regarding the possibility of long-term adverse events. This aspect has now been more comprehensively addressed in the Discussion section.

5. The information is missing how the double mutations were confirmed in cis position in the plasmid before the transfection in Hela Cells.

We thank the Reviewer for raising this question. We confirmed the double mutations were in cis position by whole plasmid sequencing performed by a third party (Eurofins Genomics). The plasmid map and the sequencing results are available in Reviewers Figure 1.

Referee #3 (Comments on Novelty/Model System for Author): The model used is appropriate for the present functional studies. However, a greater characterization to validate the lack of mTORC1 recruitment would be a useful addition.

We thank the Referee for raising this important point. As the Reviewer will see below, we have characterized more in depth the molecular mechanisms affecting the PIK3 α pathway following *PIK3R1* mutation.

Referee #3 (Remarks for Author):

The authors describe three cases of PIK3R1-related overgrowth and PI3K α inhibition with alpelisib in both the clinic and in vitro. This report adds both the number of cases of PIK3R1-related overgrowth in the literature and to the previous anecdote of alpelisib in PIK3R1 disorders. The preclinical assays provide interesting observations of an mTORC1 independent pathology that is possibly translated in the lack of sirolimus efficacy in the clinic.

Pg4 Line 11 - "The first two domains bind to the phosphatase and tensin homologue deleted on chromosome 10". What is meant by "deleted on chromosome 10"?

We apologize for this typo. We have corrected using the appropriate denomination.

Pg10 Line 18 - The data presented include a lack of pS6RP but not mTORC1 directly, thus, absence of mTORC1 is inferred by lack of pS6RP. This should be made clearer in the manuscript. Did the authors consider evaluating other elements of the AKT/mTOR signaling such as PRAS40.

In order to further support our findings, we carried out additional experiments. To this end, we transfected HeLa cells with the different variants and investigated the AKT substrates using Western blot analysis. We observed that the 3 variants produced a similar pattern of phosphorylated proteins, indicating the activation of downstream AKT targets (**new Figure Expanded View 3A and 3B**). Among them, a ~40kDa protein retained our attention because its phosphorylation levels increased in cells transfected with all three variants and decreased upon treatment with alpelisib. Given its molecular weight, we hypothesized that this protein was most likely PRAS40, a canonical effector of AKT. We confirmed that the phosphorylation levels of PRAS40 were significantly increased in cells transfected with PIK3R1 variants compared to wild-type PIK3R1, and returned to control levels upon treatment with alpelisib (**Figure Expanded View 3A and 3B**). However, we did not observe any changes in the phosphorylation levels of mTORC1 targets p70S6K and 4eBP1 (**Figure Expanded View 3A and 3B**) regardless of the variant transfected, which further supports the absence of activation of the mTORC1 pathway.

Altogether, these findings support that mTOR inhibitors may not be the most appropriate drugs for patients with *PIK3R1*-related disorders. These new data have been added to the revised version of the manuscript.

Pg11 Line 20 - Were other markers DIC such as fibrinogen, coagulopathy, thrombocytopenia abnormal at baseline? If so, did they also change during treatment with alpelisib?

As requested, we have added these new data in the revised version (**new Figure Expanded View 4A and 4B**).

Pg12 Line 8 - "The patient did not report adverse events." Adverse events should not only be the patient report but the laboratory monitoring (e.g., hyperglycemia) and clinician observations (e.g., rash). It would be helpful to specifically include what type of monitoring patients received. In addition, it is unlikely that there were no adverse events in three individuals over a 12-month period. It may, however, be possible that no drug-related adverse events occurred. Please clarify how adverse events are evaluated and reported.

The Reviewer is correct, no drug-related adverse were reported during the follow up. Patients were followed monthly during the first 3 months and then every 3 months. During their medical consultation, clinical examination, biological tests and photography were performed. MRI exams were assed prior to alpelisib introduction and then 6 and 12 months later. Biological exams include complete blood count, D-Dimers, hemostasis parameters, blood electrolytes, kidney and liver functions, glycated hemoglobin, cholesterol and triglycerides measurement. No anomalies were observed during the 12 months of follow up. Similarly, we did not observe skin rash and patients did not report diarrhea. These findings are consistent with the EPIK P1 data showing an acceptable safety profile using a low dose of alpelisib. These data are not in the revised version of the manuscript.

"Thanks to real-world evidence from the EPIK P1 clinical trial (NCT04285723), alpelisib has recently received accelerated approval from the US FDA for patients with PROS (age >2 years)(Canaud et al., 2023b, Canaud et al., 2023a). Alpelisib is associated with adverse events such as hyperglycemia, alopecia, diarrhea, and vomiting. However, in the EPIK P1 clinical trial, adverse events were less severe and less frequent, likely because of the lower dose used clinical trials(Andre et al., 2019). The most common adverse events in the EPIK P1 trial were grade 1 alopecia (16.7%), diarrhea (15.8%), hyperglycemia (12.3%), and aphthous ulcers (10.5%)⁵. It is important to note that the doses of alpelisib used here are based on real-world data, as no pharmacokinetics analyses were available."

Pg 12 - Are the authors able to comment on the trajectory of pain and fatigue improvement? For example, over the first 6 months, was their rapid enduring improvement in these symptoms or gradual linear improvement. Were formal assessments of pain and symptoms only at 6 and 12 months? The authors use the word 'rapid' multiple times in the manuscript but do not provide data on short interval follow up to support this except for the 1-month opioid follow up in patient 3.

Pain and fatigue were two symptoms that improved within the first few weeks on treatment (the reason why we called that rapid). The improvement was gradual. In order to better highlight this, we have added the graduation at 1, 2 and 3 months following alpelisib initiation. Both were assessed using the CTCAE V4 classification. These data are consistent with clinical improvement observed in PROS patients.

Pg13 Line 9 - How is pain assessed? Was the accompanied question using the VAS

for pain asked at time of clinic visit? Greatest pain in past 24 hours? Greatest pain in past week? Or were multiple questions asked? There is discrepancy in patient 3's pain being specified as a 'daily peak' but patient 1 as a weekly peak, and in patient 2 no peak was provided?

Pain was assessed at every consultation (clinic visit) using a visual analogic scale. All patients at baseline had chronic pain with additional peak of pain in two out of three. We have better clarified this information in the revised version of the manuscript.

Pg14 Line 3 - Given only three points of data pre and three post, providing an interquartile range is not particularly informative. A simple list in the text or a table would be more appropriate.

As requested, we have removed the interquartile range.

Pg15 - I would recommend the authors discuss more about the limitations of this study. This may include discussion on the volumetric analysis, lack of data during rapamycin treatment (MRI volumetrics are not presented during sirolimus treatment), and other considerations.

As requested by the Reviewer, we have now expanded the Discussion section particularly focusing on the limit of the study.

Unfortunately, we do not have MRI of these patients while they were on rapamycin. In fact, these 3 patients were followed in 3 distinct French centers and treated empirically by rapamycin for a supposed PIK3CA-related overgrowth spectrum disorder. We were solicited in the 3 cases because of rapamycin inefficacy. We then identified the PIK3R1 mutation in the 3 patients and started this work.

“This study has several limitations. Indeed, we do not have proper volumetric evaluation of the malformations using MRI during the rapamycin treatment period for the three patients to assess disease progression on this medication. Furthermore, the low number of patients in this case series requires to be cautious before to draw any conclusion. Finally, the variation in term of volumetric response to rapamycin will deserve further investigations including pharmacokinetics.”

Figure 1 A - CD31 is listed in the figure but is not included as a primary antibody in the methods section. Additionally, there does not appear to be any CD31 positive cells in the figure. Do the authors have an explanation for why the few pS6RP positive cells are not also CD31 positive?

We do apologize this is a mistake, no CD31 staining has been shown in this picture.

REFERENCES

- ANDRE, F., CIRUELOS, E., RUBOVSKY, G., CAMPONE, M., LOIBL, S., RUGO, H. S., IWATA, H., CONTE, P., MAYER, I. A., KAUFMAN, B., YAMASHITA, T., LU, Y. S., INOUE, K., TAKAHASHI, M., PAPAI, Z., LONGIN, A. S., MILLS, D., WILKE, C., HIRAWAT, S., JURIC, D. & GROUP, S.-S. 2019. Alpelisib for PIK3CA-Mutated, Hormone Receptor-Positive Advanced Breast Cancer. *N Engl J Med*, 380, 1929-1940.
- CANAUD, G., GUTIERREZ, J. C. L., IRVINE, A. D., VABRES, P., HANSFORD, J. R., ANKRAH, N., BRANLE, F., PAPADIMITRIOU, A., RIDOLFI, A., O'CONNELL, P., TURNER, S. & ADAMS, D. M. 2023a. Alpelisib for Treatment of Patients With PIK3CA-Related Overgrowth Spectrum (PROS). *Genet Med*, 100969.
- COTTRELL, C. E., BENDER, N. R., ZIMMERMANN, M. T., HEUSEL, J. W., CORLISS, M., EVENSON, M. J., MAGRINI, V., CORSMEIER, D. J., AVENARIUS, M., DUDLEY, J. N., JOHNSTON, J. J., LINDHURST, M. J., VIGH-CONRAD, K., DAVIES, O. M. T., COUGHLIN, C. C., FRIEDEN, I. J., TOLLEFSON, M., ZAENGLEIN, A. L., CILIBERTO, H., TOSI, L. L., SEMPLE, R. K., BIESECKER, L. G. & DROLET, B. A. 2021. Somatic PIK3R1 variation as a cause of vascular malformations and overgrowth. *Genet Med*, 23, 1882-1888.
- KUENTZ, P., ENGEL, C., LAENG, M., CHEVARIN, M., DUFFOURD, Y., MARTEL, J., PIARD, J., MORICE-PICARD, F., AUBERT, H., BESSIS, D., GUERROT, A. M., MARUANI, A., BOCCARA, O., MAZEREEUW-HAUTIER, J., OTT, H., PHAN, A., PUZENAT, E., QUELIN, C., THAUVIN-ROBINET, C., FAIVRE, L. & VABRES, P. 2024. Clinical phenotype of the PIK3R1-related vascular overgrowth syndrome. *Br J Dermatol*.
- MORIN, G. M., ZERBIB, L., KALTENBACH, S., FRAISSENON, A., BALDUCCI, E., ASNAFI, V. & CANAUD, G. 2024. PIK3CA-Related Disorders: From Disease Mechanism to Evidence-Based Treatments. *Annu Rev Genomics Hum Genet*.
- SIEGEL, D. H., COTTRELL, C. E., STREICHER, J. L., SCHILTER, K. F., BASEL, D. G., BASELGA, E., BURROWS, P. E., CILIBERTO, H. M., VIGH-CONRAD, K. A., EICHENFIELD, L. F., HOLLAND, K. E., HOGELING, M., JENSEN, J. N., KELLY, M. E., KIM, W., KING, D. M., MCCUAIG, C., MUELLER, K. A., POPE, E., POWELL, J., PRICE, H., STEINER, J. E., FRIEDEN, I. J., TOLLEFSON, M. M. & DROLET, B. A. 2018. Analyzing the Genetic Spectrum of Vascular Anomalies with Overgrowth via Cancer Genomics. *J Invest Dermatol*, 138, 957-967.

17th Jul 2024

Dear Guillaume,

Thank you for submitting your revised study, and please accept my apologies for the delay in getting back to you as one referee needed more time to provide his/her report. We have now received the feedback from the 3 initial referees, and as you will see below, while referees #2 and #3 are satisfied with the revisions, referee #1 still has some remaining concerns. These do not require extensive experiments, and we would like you to address them in a minor round of revisions. Additionally, please address the following editorial issues:

1/ Manuscript text:

- Please note that the author Estelle Balducci has not been entered in the submission system.
- Please indicate in track changes mode any new modification.
- Please provide up to 5 keywords.
- Your text was cross-checked for similarities with other manuscripts, and a resemblance was found with one of your previous manuscript (page 13, last paragraph). Please modify your text accordingly (see the parts of your text highlighted in the attached document).
- Methods:
 - o All Materials and Methods need to be described in the main text using our 'Structured Methods' format, which is now required for all research articles. According to this format, the Methods section includes a Reagents and Tools Table (listing key reagents, experimental models, software and relevant equipment and including their sources and relevant identifiers) followed by a Methods and Protocols section describing the methods using a step-by-step protocol format. The aim is to facilitate adoption of the methodologies across labs. More information on how to adhere to this format as well as a downloadable template (.docx) for the Reagents and Tools Table can be found in our author guidelines:
<https://www.embopress.org/page/journal/17574684/authorguide#structuredmethods>
 - o Cells: please indicate the origin of the cells, and whether they were authenticated and tested for mycoplasma contamination.
 - o Antibodies: please indicate dilutions/concentrations
 - o Patients: please include a statement that the experiments conformed to the principles set out in the WMA Declaration of Helsinki and the Department of Health and Human Services Belmont Report.
 - o Statistical analysis: please include a statement on blinding, randomization, sample size, inclusion/exclusion criteria (see checklist).
- Acknowledgements: the funding information provided in this section should match the information entered in the submission system (currently, the following needs to be entered in the submission system via separate Funder entries: CLOVES SYNDROME COMMUNITY (West Kennebunk, USA), Association Syndrome de CLOVES (Nantes, France), Fondation d'entreprise IRCEM (Roubaix, France), Fonds de dotation Emmanuel BOUSSARD (Paris, France), MSD Avenir - grant Signalopathies (Paris, France), the Fondation DAY SOLVAY (Paris, France), MSD Avenir (Signalopathies grant), the Fondation TOURRE (Paris, France) to GC, the Fondation BETTENCOURT SCHUELLER (Paris, France) to GC, the Fondation Simone et Cino DEL DUCA (Paris, France), the Fondation Line RENAUD-Loulou GASTE (Paris, France), the Fondation Schlumberger pour l'Education et la Recherche (Paris, France), the Fondation Maladies Rares, the Association Robert Debré pour la Recherche Médicale, WonderFIL smiles - a Facial Infiltrating Lipomatosis community (Norway), INSERM, Assistance Publique Hôpitaux de Paris, l'Université Paris Cité, Fondation pour la Recherche Médicale (FDM202006011222) and Banting Postdoctoral Fellowship (Canadian Institutes of Health Research, #472149)
- Author contributions: CRediT has replaced the traditional author contributions section because it offers a systematic machine readable author contributions format that allows for more effective research assessment. Please remove the Authors Contributions from the manuscript and use the free text boxes beneath each contributing author's name in our system to add specific details on the author's contribution.
- "Conflict of interest" should be renamed "Disclosure statement and competing interests". Please include the sentence "G. Canaud is an editorial advisory board member".
- Data availability section: please replace the current sentence by: "This study includes no data deposited in external repositories." This section should be after "Methods" and before "Acknowledgements".
- References: please reformat the references with "et al" after 10 author names, and year of publication.

2/ Figures:

- Please upload the figures individually.
 - All figures and figure panels should be referenced in the text. Currently, Supplementary Fig. 4A-B needs to be updated to the correct figure; Figure 2 has only panels A-C, but the following panels are also called out in the manuscript: 2E, 2F, 2I, 2J.
 - Supplementary Table 1 should be renamed Table EV1 (callouts included).
 - Please address the queries from our data editors in the figure legends:
- Please note that the exact p values are not provided in the legends of figures 1b-c; 2c; 3a; EV 1.

3/ Source Data:

Thank you for providing the raw data underlying your figures. Please upload the SD for the main figures as 1 folder per figure. The SD for EV figures can be grouped into one single folder.

4/ Please provide a complete author checklist, which you can download from our author guidelines

(<https://www.embopress.org/page/journal/17574684/authorguide#submissionofrevisions>). Please insert information in the checklist that is also reflected in the manuscript. The completed author checklist will also be part of the RPF.

5/ Synopsis:

- text: please include a short stand first before your bullet points (maximum 300 characters, including space), remove the synopsis text from the manuscript and upload it separately.

- image: please update the format to jpeg, tiff or png, 550 pixels wide x 200-600 pixels wide

6/ As part of the EMBO Publications transparent editorial process initiative (see our Editorial at

<http://embomolmed.embopress.org/content/2/9/329>), EMBO Molecular Medicine will publish online a Review Process File (RPF) to accompany accepted manuscripts.

This file will be published in conjunction with your paper and will include the anonymous referee reports, your point-by-point response and all pertinent correspondence relating to the manuscript. Let us know whether you agree with the publication of the RPF.

I look forward to receiving your revised manuscript.

With kind regards,

Lise

***** Reviewer's comments *****

Referee #1 (Remarks for Author):

While the manuscript has improved, and I still have some comments which should be addressed.

Are sections shown in Fig. 1A from overgrowth tissue? Please explain.

From the images provided it is not clear that pAKT levels are higher in carriers of PIK3R1, at least not in vessels. Which cell types are you quantifying? Would it not be better to quantify intensity to claim increase pAKT levels?

I do not understand how authors have done the quantifications of Fig. 1B. First, are they using the same controls for c1735_1740del and for c1372_1373dupAAA. If so, these data should be represented in the same graph, and apply the corresponding statistical test (multivariate test). It is interesting that they seem to use the same controls when graphing p-S6RP but different ones when showing pAKT. Why is that? The same applies to western blot in Fig. 3

In line 16, authors still use recruitment instead of activation. Please correct.

p110a antibodies are very tricky for IF, and not selective for p110a Control samples of p110a KO cells showing that the antibody shows no immunoreactivity must be shown.

Referee #2 (Remarks for Author):

The authors responded to all of my comments

Referee #3 (Remarks for Author):

The authors have responded to reviewer comments with additional experiments and details.

Referee #1 (Remarks for Author):

While the manuscript has improved, and I still have some comments which should be addressed.

Are sections shown in Fig. 1A from overgrowth tissue? Please explain. From the images provided it is not clear that pAKT levels are higher in carriers of PIK3R1, at least not in vessels. Which cell types are you quantifying? Would it not be better to quantify intensity to claim increase pAKT levels?

Yes, biopsies were performed on the affected tissues, as evidenced by the identification of the PIK3R1 mutation in these samples.

As previously mentioned during the first revision, the quantification of immunofluorescence was performed as follows:

“P-S6RP and P-AKT^{T308} signal in human skin samples were analyzed with ImageJ (v1.54f). After background subtraction, an automatic threshold was applied to each image to discriminate positive and negative areas (<https://imagej.net/plugins/auto-threshold#Moments>). To obtain a better assessment of AKT and S6RP phosphorylation in blood vessels and soft tissues, areas containing skin structures interfering with staining analysis (epidermis and hair follicles) were excluded by hand-contouring. P-AKT and P-S6RP-positive surface areas were assessed by dividing the positive surface by the total surface analyzed in each image and expressed as percentages. The figure pictures were selected among representative images of the statistical results. As control, we used 3 biopsies from 3 healthy volunteers.”

We chose to analyze the surface area of tissue stained for AKT and S6RP, excluding hair follicles and epidermis, rather than measuring intensity in endothelial cells. This decision was based on the observation that patients exhibited overgrowth and vascular malformations, suggesting that the mutation may be present in cell types other than endothelial cells. This has been clarified in the revised version of the manuscript.

I do not understand how authors have done the quantifications of Fig. 1B. First, are they using the same controls for c1735_1740del and for c1372_1373dupAAA. If so, these data should be represented in the same graph, and apply the corresponding statistical test (multivariate test). It is interesting that they seem to use the same controls when graphing p-S6RP but different ones when showing pAKT. Why is that? The same applies to western blot in Fig. 3.

Originally, the data for the c1735_1740del and c1372_1373dupAAA variants were graphed separately, despite being associated with the same control datasets. When displaying P-AKT, the exact same control data were used; however, differences in the y-axis scales for each variant may have created the illusion of variation. To adhere to best statistical practices and in response to the referee's request, the c1735_1740del and c1372_1373dupAAA variants were consolidated into the same graph in Figures 1 and 3, and the appropriate multivariate test was applied in the revised version of the manuscript.

In line 16, authors still use recruitment instead of activation. Please correct.

We apologize and we have modified accordingly.

p110a antibodies are very tricky for IF, and not selective for p110a Control samples of p110a KO cells showing that the antibody shows no immunoreactivity must be shown.

We agree with the Reviewer that p110 α antibodies are challenging to use for IF. We would like to emphasize that this experiment was performed during the previous revision at the Reviewer's request. We spent eight months attempting to validate the specificity of this antibody. To achieve this, we transfected HeLa cells with three different PIK3CA siRNAs. Although we obtained approximately an 80% reduction in p110 α expression, the remaining ~20% expression did not allow us to formally determine the antibody's specificity at the IF level (**Reviewer Figure 1**). Indeed, we cannot confirm the specificity of the antibody at the IF level. If the Editors want, we can remove this specific part, which was not initially included in the original version.

Reviewer Figure 1:

Knock-Down efficiency as assessed by Western Blot

Referee #2 (Remarks for Author):

The authors responded to all of my comments

Referee #3 (Remarks for Author):

The authors have responded to reviewer comments with additional experiments and details.

24th Apr 2025

Dear Guillaume,

Thank you for submitting your revised files. As you will see below, referee #1 is satisfied with the revisions, and I am pleased to inform you that your manuscript is now accepted for publication!

Before we can send it to our publisher, please reformat the references to include 10 authors before et al. Please also include the following full sentence in the conflict of interests section:

"Guillaume Canaud is an editorial advisory board member. This has no bearing on the editorial consideration of this article for publication."

Once these changes are made, please send me your manuscript text file via email and I'll upload it in the submission system.

The attached cropped image will serve as a thumbnail on our electronic table of content. Please let us know if you agree, or kindly provide an alternative image (115x70 pixels).

If you have any questions, please do not hesitate to contact the Editorial Office.
Thank you for your contribution to EMBO Molecular Medicine!

With kind regards,

Lise

Referee #1 (Remarks for Author):

The authors responded to all of my comments

Referee #1 (Remarks for Author):

The authors responded to all of my comments
